# `AdaSTaR`: Adaptive Data Sampling for Training Self-Taught Reasoners

**Woosung Koh[1,3], Wonbeen Oh[3], Jaein Jang[3], MinHyung Lee[3], Hyeongjin Kim[3],**
**Ah Yeon Kim[3], Joonkee Kim[2], Junghyun Lee[1], Taehyeon Kim[2]\*, Se-Young Yun[1]\***
[1]KAIST AI, [2]LG AI Research, [3]Yonsei University
{reiss.koh,yunseyoung}@kaist.ac.kr, kimtaehyeon610@gmail.com

## Abstract

Self-Taught Reasoners (`STaR`), synonymously known as Rejection sampling Fine-Tuning (`RFT`), is an integral part of the training pipeline of self-improving reasoning Language Models (LMs). The self-improving mechanism often employs random observation (data) sampling. However, this results in trained observation imbalance; inefficiently over-training on solved examples while under-training on challenging ones. In response, we introduce Adaptive `STaR` (`AdaSTaR`), a novel algorithm that rectifies this by integrating two adaptive sampling principles: (1) Adaptive Sampling for Diversity: promoting balanced training across observations, and (2) Adaptive Sampling for Curriculum: dynamically adjusting data difficulty to match the model's evolving strength. Across six benchmarks, `AdaSTaR` achieves best test accuracy in all instances (6/6) and reduces training FLOPs by an average of 58.6% against an extensive list of baselines. These improvements in performance and efficiency generalize to different pre-trained LMs and larger models, paving the way for more efficient and effective self-improving LMs.

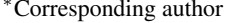 github.com/reiss-koh/AdaSTaR

## 1 Introduction

Language models (LMs) are demonstrating remarkable emergent abilities across diverse cognitive tasks such as mathematical reasoning (Yao et al., 2023; Chen and Li, 2024; Brown et al., 2024), code generation (Sun et al., 2024; Research, 2025), and commonsense reasoning (Qwen Team, 2023; Google, 2023). Although LMs acquire foundational reasoning capabilities from large-scale pre-training and supervised finetuning (SFT), generating high-quality, explicit reasoning steps, often called Chains-of-Thought (CoT) (Wei et al., 2022a, 2023, 2022b; Wang et al., 2023a), typically requires costly human annotation (Lightman et al., 2024; Havrilla et al., 2024; Zelikman et al., 2024). Creating such datasets is expensive and scales poorly, presenting a critical bottleneck as tasks increase in complexity. This challenge motivates the de-

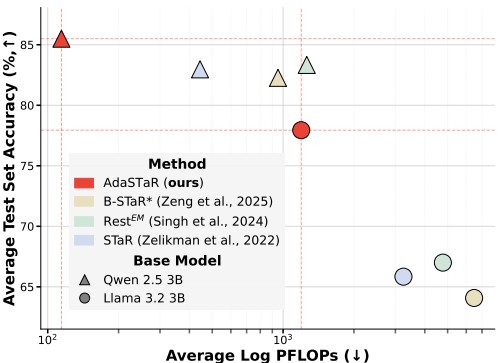

Figure 1: Average test accuracy and FLOPs across six datasets for `Llama 3.2 3B` and three datasets for `Qwen 2.5 3B`. Results consistently extend to `Gemma 7B` as well. \*We use outcome verification on `B-STaR` for fair comparison. Thus, the implementation with process verification may perform significantly better.

---

\*Corresponding author

39th Conference on Neural Information Processing Systems (NeurIPS 2025).

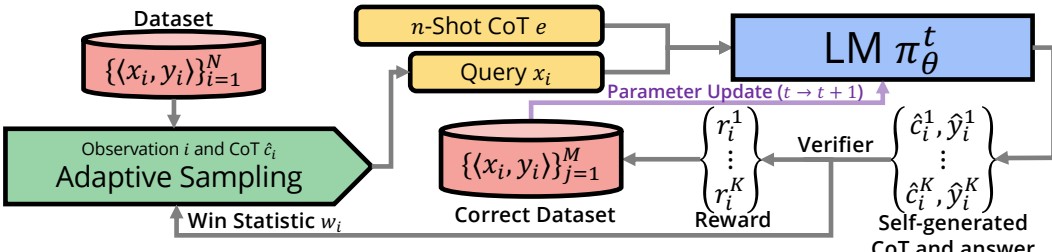

Figure 2: High-level schematic diagram of `AdaSTaR`. Other `STaR`-like approaches are equivalent to this diagram, excluding the win statistic $w_i$ computation and the Adaptive Sampling module.

velopment of methods that improve LM reasoning without relying on extensive human annotation.

Self-improvement mechanisms, such as Self-Taught Reasoners (`STaR`; Zelikman et al., 2022), also referred to as Rejection-sampling Fine-Tuning (`RFT`; Yuan et al., 2023; Singh et al., 2024), offer a promising alternative. The core idea behind `STaR` is to enable the LM to iteratively improve itself: the model generates CoTs, verifies the final answer against ground truth, and fine-tunes on CoTs that yield correct answers. This iterative inference, verify, and train cycle allows LMs to generate their own training data, circumventing the need for human-annotated CoTs.

However, while reducing annotation costs, the standard `STaR` framework, which relies on random data sampling, suffers from inefficiencies and learning challenges. The random sampling often leads to a training data imbalance: the model wastes compute repeatedly re-training on examples it can already solve, while potentially under-sampling more challenging examples where learning is most needed (Singh et al., 2024). This imbalance results in inefficient use of training compute and contributes to `STaR`'s significantly slower convergence compared to standard SFT (see Fig. 5 in Appendix §A).

Furthermore, `STaR`'s reliance on outcome verification (checking only the final answer) means it can inadvertently train on flawed or suboptimal CoTs that happen to reach the correct answer (Kawabata and Sugawara, 2024; Lee et al., 2025). Reinforcing these "false positives" can degrade the model's underlying reasoning capabilities. While Process Reward Models (`PRM`; Lightman et al., 2024; Zeng et al., 2025) that assess the CoTs can mitigate this, PRMs require their own significant annotation and computational overhead (Lu et al., 2024; Setlur et al., 2025). We therefore view PRMs as an orthogonal approach. Consequently, a key challenge in `STaR`-based self-improvement is balancing the exposure to diverse problem difficulties with the need to maintain training data quality, as sampling harder examples is more likely to yield noisy or incorrect CoTs. This leads to a research question: How can `STaR` achieve efficient and effective self-improvement by balancing diverse learning exposure while maintaining the quality of self-generated CoTs?

**Contribution.** We propose `Adaptive STaR` (`AdaSTaR`), a novel method that integrates adaptive sampling into the `STaR` training loop. `AdaSTaR` implements two core intuitions: (1) Adaptive Sampling for Diversity: prioritizing under-trained examples to ensure balanced learning; and (2) Adaptive Sampling for Curriculum: regularizing the system to sample easier data when the model is weaker early on. We empirically validate the effectiveness and efficiency of `AdaSTaR` through experiments across six reasoning datasets and an extensive list of baselines. `AdaSTaR` consistently improves both performance and computational efficiency. Remarkably, `AdaSTaR` not only achieves the highest test accuracy across **all 6/6 benchmarks**, but also simultaneously reduces the required training compute (FLOPs) by an average of **58.6%** compared to the strongest accuracy baseline (see Fig. 1). These performance and efficiency gains generalize to other pre-trained LMs and larger model size which we discuss further later.

**Related Work.** Although many works build on `STaR`, none, to our knowledge, target improving efficiency. Subsequent works improve performance at significant compute cost; `AdaSTaR` is complementary, improving scalability and accessibility. `V-STaR` (Hosseini et al., 2024) adds a verifier LM to improve inference-time performance through best-of-$N$ sampling (Snell et al., 2025). Iterative Reasoning Preference Optimization (Pang et al., 2024) incorporates a Direct Preference Optimization (Rafailov et al., 2023) term in its objective: to curate preference pairs, it increases CoT samples from

$K = 2$ in `STaR` to $K = 30$. `B-STaR` (Zeng et al., 2025) enhances LM exploration for more diverse reasoning, and trains a separate process reward model (Uesato et al., 2022; Lightman et al., 2024) for finer-grained verification. `Lean-STaR` (Lin et al., 2025) employs the Lean theorem prover (De Moura et al., 2015) and a frontier LM (GPT-4) to extend `STaR` to mathematical theorem proving.

Reinforcement Learning (RL) offers a parallel approach to enhance LM reasoning, also leveraging an iterative process. RL's reward-based objective often yields long-CoTs (Shao et al., 2024; DeepSeek-AI, 2025; Liu et al., 2025b; Yu et al., 2025b; Sui et al., 2025; Kimi Team, 2025; Liu et al., 2025a; Yu et al., 2025a), unlike the short-CoTs (NVIDIA, 2025) typical of `STaR`-style SFT. While the significantly larger token generation size of RL-based long-CoTs result in top performers, integrating `STaR`'s SFT remain a salient part of the training pipeline (Sui et al., 2025). For instance, Kimi k1.5 (Kimi Team, 2025), a representative reasoning model, utilizes `STaR` to expand their primary SFT dataset. To address the difficult, mixed-language, and overly long CoTs, DeepSeek-R1 (DeepSeek-AI, 2025) and Kimi k1.5 incorporate a `STaR` stage. Finally, DeepSeek-GRM (Liu et al., 2025b), a generalist reward model, also adopts a modified `STaR` as its training's first stage. While these RL-based advancements are significant, our work concentrates on enhancing the `STaR` stage.

## 2 Preliminary and Motivation

### 2.1 Preliminary: Self-Taught Reasoner (`STaR`) and its Variants

Let $\pi_\theta^t$ denote a LM (Vaswani et al., 2017) parameterized by $\theta$ at iteration $t$. We are given a supervised dataset $\mathcal{D} = \{\langle x_i, y_i \rangle\}_{i=1}^N$. Following Wei et al. (2022b), each task is represented as $\langle x, c, y \rangle$, where $x \in \mathcal{X}$ is the query (input), $c \in \mathcal{C}$ is the CoT reasoning step(s), and $y \in \mathcal{Y}$ is the final answer. Since ground-truth CoTs $\mathcal{C}$ are unavailable, `STaR` aims to generate appropriate $c$ to improve generalization. To achieve this, $\pi_\theta^t$ generates $\langle \hat{c}_i, \hat{y}_i \rangle$ conditioned on fixed few-shot CoT exemplars $e = \{\langle x_\epsilon, c_\epsilon, y_\epsilon \rangle\}_{\epsilon=1}^E$. However, as no ground truth $c_i$ is available, we require sampling and verification. Given the supervised dataset, a rule-based verifier defines a reward signal $r := \mathbb{I}(y_i = \hat{y}_i)$, where $\mathbb{I}(\cdot)$ is the indicator function.

Let $K \in \mathbb{N}$ denote the number of CoT traces sampled as follows (Fig. 2, blue). For the first $k \in \{1, 2, \cdots, K\}$, each observation $i$ is sampled once via $\langle \hat{c}_i, \hat{y}_i \rangle \leftarrow \pi_\theta^t(e, x_i)$. If $r = 1$, it is accepted, and if $r = 0$, it is resampled using rationalization (Zelikman et al., 2022): $\pi_\theta^t(e, x_i \oplus y_i)$, where the ground truth $y_i$ is concatenated. In some extensions of `STaR`, $K > 2$ samples are drawn without rationalization (Singh et al., 2024; Hosseini et al., 2024; Pang et al., 2024; Zeng et al., 2025; Lin et al., 2025).

Correct samples $\mathcal{D}_+^t := \{\langle x_i, \hat{c}_i, \hat{y}_i \rangle | y_i = \hat{y}_i\}$ are re-random-sampled down to match the per-iteration batch size $\beta^t = \sigma^t \cdot \beta$, then used for negative log-likelihood (NLL) learning. Here, the step size $\sigma^t$ is the number of parameter updates per iteration $t$. Here all superscript $t$ indicates iteration, not a numerical exponent operation. Initial $\beta^{t=1} = 40 \cdot 8 = 320$ as presented in the original implementation (Zelikman et al., 2022). $\beta^t$ rises over time as we follow $\beta^{t+1} := 1.2(\beta^t)$ in the original implementation. However, alternative `STaR`-based approaches (Hosseini et al., 2024; Pang et al., 2024; Zeng et al., 2025; Lin et al., 2025; Peng et al., 2025) remove this pre-determined $\beta^t$, and instead set $\beta^t$ to $|\mathcal{D}_+^t|$.

Post gradient updates, $\pi_\theta^t$ transitions to $\pi_\theta^{t+1}$ (Fig. 2, purple). Two inter-iteration strategies exist across `STaR`-based methods: (1) resetting: always retrain from the base model: $\pi_\theta^{t+1} \leftarrow \text{Train}(\pi_\theta^{t=1}, \mathcal{D}_+^t)$ (Zelikman et al., 2022; Hosseini et al., 2024; Singh et al., 2024); (2) accumulating: incrementally fine-tune from the previous model: $\pi_\theta^{t+1} \leftarrow \text{Train}(\pi_\theta^t, \mathcal{D}_+^t)$ (Pang et al., 2024; Zeng et al., 2025; Lin et al., 2025; Peng et al., 2025).

### 2.2 Motivation: Need for Adaptive Data Sampling

**`STaR`'s data sampling induces persistent inefficient imbalance in training data.** A key finding is that `STaR`'s sampling strategy leads to some observations being over-trained while others are under-trained. This training frequency imbalance is empirically illustrated in Fig. 3a. The pattern of variance in observation training frequency is persistent across all datasets examined (see Appendix § B for all visualizations). As the filtered set $\mathcal{D}_+^t$ consists exclusively of observations for which the LM correctly produced $\hat{y}_i$, a high variance naturally arises in how often each distinct observation $i$ is

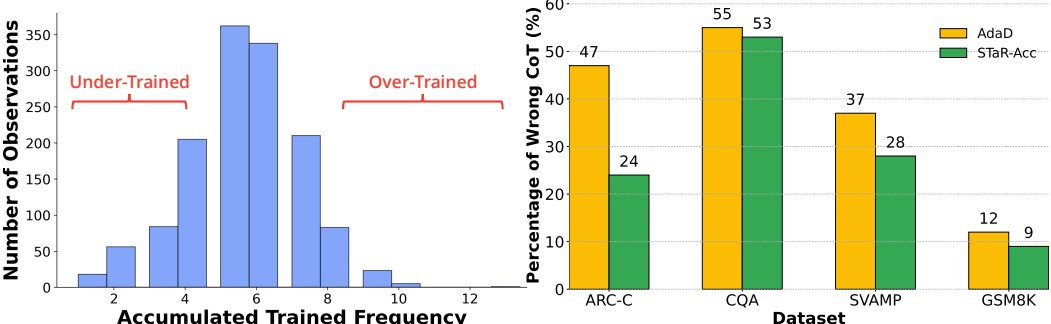

(a) Distribution of frequency trained of each observation $i$ in iterations 1 to 10; in ARC-C.

(b) Percentage of wrong CoT when the answer is correct $(y_i = \hat{y}_i)$ for `AdaD` and `STaR-Acc`.

Figure 3: Empirical motivation for the need for adaptive sampling of diverse observations (a), regularized with curriculum learning (b).

trained. Consequently, more challenging observations (left tail of Fig. 3a) are included in $\mathcal{D}_+^t$ less frequently and become under-trained, whereas easier ones (right tail) are over-represented and thus over-trained. In this example, challenging observations are trained 1–2 times, while easier ones are trained 10–13 times. This results in inefficient compute usage, as resources are repeatedly spent on observations that the model $\pi_\theta^t$ already solves reliably. This situation mirrors the motivation for early stopping in LM training, which aims to avoid overfitting to already-solved data (Caruana et al., 2000; Kaplan et al., 2020; Hernandez et al., 2021).

We further examine whether observations initially under- or over-trained remain in these respective regimes over iterations. Empirically, even after three iterations $(t + 3)$, averaging across six datasets, 72.4% of observations initially in the first quartile (Q1) of training frequency remain in Q1, and 91.2% of observations from the fourth quartile (Q4) remain in Q4. Detailed visualizations are available in Appendix § B. This suggests the phenomenon is chronic and does not self-alleviate without intervention.

**Prioritizing harder examples for diversity elevates false positives, motivating curriculum-based regularization.** However, encouraging training diversity by biasing sampling toward harder observations (left tail of Fig. 3a) can increase false positives. False positives are defined as cases where the predicted answer $\hat{y}$ is correct but the generated CoT $\hat{c}$ is flawed (Singh et al., 2024; Kawabata and Sugawara, 2024; Lee et al., 2025). We empirically observe that sampling more challenging observations leads to poorer quality CoTs.

Following Wei et al. (2025) and Lee et al. (2025), we leverage the strongest available teacher model (Ho et al., 2023) (GPT 4o) to annotate false positives. We compare a method encouraging diversity by sampling challenging observations (`AdaD`) against its baseline, `STaR-ACC`. `AdaD` and `STaR-Acc` are formally introduced in § 3 and 4.1. For each method, 100 observations are randomly sampled (without replacement) from $\mathcal{D}_+^t$ for CoT annotation. The precise iteration $t$ for both methods is chosen by taking $t := \min(\text{BestIter}(\text{AdaD}), \text{BestIter}(\text{STaR-Acc}))$, where $\text{BestIter}(\cdot)$ is the early-stopped iteration. Further details and a qualitative example are provided in Appendix § C.

Fig. 3b illustrates that inducing increased training diversity can degrade CoT quality, measured by the rate of false positives across four datasets. On average, sampling more diverse and challenging observations lead to a 9% increase in false positives. Hence, we propose to *regularize* for model strength to reduce the adverse effects of sampling diverse and challenging observations. To this end, our observation sampling algorithm adopts a curriculum learning style approach (Xu et al., 2020; Wettig et al., 2024).

## 3   Method: `AdaSTaR`

This section presents `AdaSTaR`, an adaptive sampling algorithm designed to address the problems highlighted in § 2.2. Alg. 1 presents the pseudocode, where lines unique to `AdaSTaR` are highlighted in green; the remaining lines follow standard `STaR` conventions. `AdaSTaR` incorporates two mechanisms: Adaptive Data Sampling for Diversity (`AdaD`) and Adaptive Data Sampling for Curriculum (`AdaC`).

## 3.1 Adaptive Data Sampling for Diversity

**Diversity Statistic.** We track two statistics for each observation $i$: the last iteration it was sampled, $\tilde{t}_i \in \mathbb{N}_0$, and a win statistic, $w_i \in [0, 1]$. Prioritizing observations with smaller $\tilde{t}_i$ values directly promotes sampling diversity. We use the last *sampled* iteration rather than the last *trained* iteration because prioritizing based on training can cause the system to repeatedly attempt difficult examples it cannot yet solve, particularly when the model is weak, early in training. Among observations with identical $\tilde{t}_i$ values, we prioritize those deemed more difficult. This approach is reminiscent of difficulty-aware methods successful in various machine learning scenarios, such as contrastive learning (Robinson et al., 2021), active learning (Xie et al., 2021), and dataset pruning (Zheng et al., 2023; Maharana et al., 2024; Cho et al., 2025). A key contribution of AdaSTaR is its computationally efficient method for estimating observation difficulty within STaR systems.

We estimate difficulty using the win statistic $w_i$, which is computed based on model performance at $\tilde{t}_i$ (the last iteration $i$ was sampled): $w_i \equiv w_i^{\tilde{t}_i} := \frac{1}{K}\sum_{k=1}^{K}\mathbb{I}[y_i = \hat{y}_i]$, where $\hat{y}_i$ is from $\pi_\theta^{\tilde{t}_i}(e, x_i)$. This represents the proportion of correct answers out of $K$ CoT samples generated at iteration $\tilde{t}_i$. Next, we elaborate on why this is a sensible proxy for difficulty.

At each iteration $t$, we want our model to maximize $p_i^t := \mathbb{P}(y_i = \hat{y}_i \leftarrow \pi_\theta^t(x_i))$ for all $i$'s. As the model is fitted with likelihood maximization (Fisher, 1922), we can expect $p_i^{t+1} \geq p_i^t$ for any $i$ sampled at iteration $t$. It is therefore sensible to prioritize observations with the lowest $p_i^t$ values, as they require more sampling and can be interpreted as more difficult at iteration $t$. It now remains to approximate $p_i^t$. A direct Monte Carlo estimate with $K$ samples gives $p_i^t \approx \hat{p}_i^t := \frac{1}{K}\sum_{k=1}^{K}\mathbb{I}[y_i = \hat{y}_i \leftarrow \pi_\theta^t(x_i)]$. However, computing this for every $i$ at every iteration $t$ requires $K$ forward passes per observation, which is computationally prohibitive.

Instead, we reuse the most recent estimate $\hat{p}_i^{\tilde{t}_i}$. The win static computation at $\tilde{t}_i$ induces no (runtime) compute overhead as the $K$ samples are an inherent part of the existing STaR system. Recalling that $\tilde{t}_i < t$ refers to the last iteration in

---

**Algorithm 1:** AdaSTaR

**Input:** $\mathcal{D}, \pi_\theta^{t=1}, e$
/* AdaD (§3.1; lines 1-14)          */
1  $\tilde{t} \leftarrow \text{dict}\{i : \tilde{t}_i = 0\}_{i=1}^N$ ;
2  $w \leftarrow \text{dict}\{i : w_i = 0\}_{i=1}^N$ ;
3  init $\text{HieMinHeap}(\mathcal{D}, \tilde{t}, w)$ ;
4  **for** iteration $t = 1, \cdots$ **do**
5     $\mathcal{D}_+^t \leftarrow \emptyset, m \leftarrow 0$ ;
6     $w^{tmp} \leftarrow \text{dict}\{i : w_i^{tmp} = 0\}_{i=1}^N$ ;
7     **while** $|\mathcal{D}_+^t| < \beta^t$ **do**
8        $i \leftarrow \text{HieMinHeap}.peek\_next$ ;
9        $m \leftarrow m + 1$ ;
10       **for** sample $k = 1, \cdots, K$ **do**
11          $\langle \hat{c}_i, \hat{y}_i \rangle \leftarrow \pi_\theta^t(e, x_i)$;
12          $w_i^{tmp} \leftarrow \frac{k-1}{k}w_i^{tmp} + \frac{1}{k}\mathbb{I}[\hat{y}_i = y_i]$;
13          **if** $\hat{y}_i = y_i$ **then**
14             $\mathcal{D}_+^t \leftarrow \mathcal{D}_+^t \cup \{\langle x_i, \hat{c}_i, \hat{y}_i \rangle\}$ ;

/* AdaC (§3.2; lines 15-19)   */
15    $\alpha, \pi_\theta^{t+1} \leftarrow \text{Train}(\pi_\theta^t, \mathcal{D}_+^t)$ ;
16    **for** $1, \cdots, \lfloor m\alpha^2 \rfloor$ **do**
17       $i \leftarrow \text{HieMinHeap}.pop$ ;
18       $\tilde{t}_i \leftarrow t, w_i \leftarrow w_i^{tmp}$;
19       $\text{HieMinHeap}.push(i, \tilde{t}_i, w_i)$ ;

---

which $i$ was sampled, $\hat{p}_i^{\tilde{t}_i}$ is the most recently available approximation to $\hat{p}_i^t$. Moreover, as we are priority-sorting with respect to $\tilde{t}_i$, we can expect that $t - \tilde{t}_i$ is not too large, i.e., $\hat{p}_i^t \approx \hat{p}_i^{\tilde{t}_i}$.

**Implementation.** As input, AdaSTaR takes the original dataset $\mathcal{D}$, base model $\pi_\theta^{t=1}$, and $n$-Shot CoT examplar $e$. For all observations, the statistics are initialized to 0 (lines 1, 2). In line 3, we utilize Cormen et al. (2022)'s Hierarchical Min Heap HieMinHeap to order the observations via the two statistics as follows: for two observations $i, j \in \text{HieMinHeap}(\cdot, \tilde{t}, w)$,

$$\underbrace{i \succ j}_{i \text{ is peeked/popped before } j} \iff \underbrace{\tilde{t}_i < \tilde{t}_j}_{i \text{ is last sampled before } j} \vee \underbrace{(\tilde{t}_i = \tilde{t}_j \wedge w_i < w_j)}_{\substack{i \text{ and } j \text{ are last sampled at the} \\ \text{same } t, \text{ but } i \text{ is more difficult}}}. \tag{1}$$

For each iteration $t$, a new empty $\mathcal{D}_+^t$ is initialized (line 5), which is used for the training at the end (line 15). We also initialize $m := 0$, which counts the number of sampled observations (line 9), and $w^{tmp}$, a dictionary of computed win-rates at iteration $t$ (line 12). The **while** loop sequentially samples $i$ from HieMinHeap, then updates the win-rate $w_i^{tmp}$ over $K$ samples of CoT-answer pairs $\langle \hat{c}_i, \hat{y}_i \rangle$ (lines 11-12) and adds $\langle x_i, \hat{c}_i, \hat{y}_i \rangle$ to $\mathcal{D}_+^t$ if $\hat{y}_i$ is correct (lines 13-14).

**Remark 1** (Non-excessive sampling in line 7). *The* **while** *loop terminates once* $|\mathcal{D}_+^t| \geq \beta^t$*. This avoids overhead from exhaustively sampling all observations before pruning to* $\beta^t$*, a practice in some prior* `STaR` *implementations (see Appendix § D for further discussion).*

### 3.2 Adaptive Data Sampling for Curriculum

To avoid over-sampling challenging observations ($\downarrow \tilde{t}_i, \downarrow w_i$) when the model is weak, we regularize `AdaD` using an adaptive curriculum. A natural approach is to incorporate curriculum learning (Hacohen and Weinshall, 2019; Kong et al., 2021) by mixing easier observations when the model is weak, then gradually reducing their ratio as it improves. This strategy aligns with curriculum learning for LM training (Pouransari et al., 2024; Li et al., 2024; Zhao et al., 2025) and is supported by data selection literature showing that combining easy and hard samples yields better outcomes than selecting only hard samples (Zheng et al., 2023; Maharana et al., 2024; Cho et al., 2025).

We use the training accuracy $\alpha \in [0, 1]$ from the current iteration $t$ as a proxy for model strength (Alg. 1, line 15). When $\alpha$ is low (indicating a weaker model), a relatively easier mix of observations should be prioritized for subsequent sampling. This regularization is automatically phased out as $\alpha$ increases with training. Similar to tracking $\tilde{t}_i$ and $w_i$, using $\alpha$ introduces no additional computational overhead, as the training step (which yields $\alpha$) is integral to the system. This explains our choice over, for instance validation set accuracy (not used in final evaluation); while potentially a more robust measures of generalization, these would require additional inference passes not intrinsic to the `STaR` loop.

**Implementation.** The curriculum component (Alg. 1, lines 15-19) implements a curriculum by adjusting statistic-update frequency based on model strength $\alpha$. Of the $m$ sampled observations per iteration, only the $\lfloor m\alpha^2 \rfloor$ highest-priority ones are popped; their statistics are updated ($\tilde{t}_i \leftarrow t$, $w_i \leftarrow w_i^{tmp}$) before reinsertion.[2] Consequently, when $\alpha$ is low (model is weak), a larger proportion of the $m$ considered observations are not updated. These non-updated observations retain their existing statistics, increasing their re-selection likelihood in the subsequent iteration. This implicitly mixes easy observations when $\alpha$ is low, avoiding the cost of explicitly identifying and mixing them.

## 4 Experiments

### 4.1 Experimental Protocol

**Setup.** We conduct our main experiments with `Llama 3.2 3B` (Llama Team, 2024). We also evaluate using `Qwen 2.5 3B` (Qwen Team, 2024) and `Gemma 7B` (Gemma Team, 2024) to demonstrate the generality of our method across different model families. All base models are pre-trained-only models. For fairness, we optimize hyperparameters using the original `STaR` and apply them consistently across all methods. Further experimental details are provided in Appendix § E.

**Datasets.** We attempt to get a wide coverage of reasoning tasks by using six well-known datasets. We use the AI2 Reasoning Challenge's Challenge set (ARC-C; Clark et al., 2018) for scientific reasoning, CommonsenseQA (CQA; Talmor et al., 2019) for commonsense reasoning, and CLadder 1.5 (Jin et al., 2023) for causal reasoning. For natural language inference reasoning we use Adversarial NLI (ANLI; Nie et al., 2020). For mathematical reasoning we use GSM8K (Cobbe et al., 2021) and SVAMP (Patel et al., 2021). For the mathematical reasoning datasets, we disable rationalization (i.e., providing hints) as it meaningfully degrades performance. Moreover, we unavoidably use `Qwen 2.5 3B` for GSM8K, as all `STaR`-based methods fail to self-improve with `Llama 3.2 3B` as the base model. We discuss this further in Appendix § F.

**Evaluation.** We use two evaluation metrics: Test Set Accuracy (Acc.) and Floating Point Operations (FLOPs). The corresponding early-stopped (Caruana et al., 2000) epoch (e) and iteration (it) for vanilla SFT and `STaR`-based approaches, respectively are reported. All methods are given an equal and large compute budget to ensure that the peak value is obtained via early-stopping. For reproducibility, we evaluate accuracy using zero-shot greedy decoding unless stated otherwise. We use FLOPs as our

---

[2] The choice of $f(\alpha) := \alpha^2$ is a hyperparameter. It allows more repetition of easy observations when the model is weak, and *rapidly* phases out this regularization effect as the model strengthens.

Table 1: Empirical results where Test Set Accuracy (%, ↑) is reported under zero-shot greedy decoding, excluding the 5-SC evaluation. Total training costs are reported in Peta FLOPs (PFLOPs, ↓). Best Acc. and PFLOPs is **bolded**, and second best is underlined in each section (excluding SFT). In (red) we quantify percent PFLOPs reduction against the highest accuracy baseline.

| Evaluation | ARC-C | | | CQA | | | CLadder 1.5 | | |
| --- | --- | --- | --- | --- | --- | --- | --- | --- | --- |
| Metric | Acc. (↑) | $t$ | PFLOPs (↓) | Acc. (↑) | $t$ | PFLOPs (↓) | Acc. (↑) | $t$ | PFLOPs (↓) |
| SFT | 61.4 | 1.0 e | 7.0 | 71.8 | 1.0 e | 24.0 | 31.0 | 7.0 e | 382.3 |
| SFT + 8-CoT | 59.0 | 1.5 e | 10.5 | 71.6 | 2.5 e | 60.1 | 43.6 | 3.0 e | 163.9 |
| SFT + 5-SC | 63.8 | 4.5 e | 31.6 | 76.4 | 2.5 e | 60.1 | 45.2 | 8.0 e | 437.0 |
| STaR | 71.6 | 13 it | 351.4 | 72.2 | 25 it | 2877.8 | 53.4 | 25 it | 8427.3 |
| STaR-Full | 69.8 | 27 it | 739.4 | 72.2 | 12 it | 1502.7 | 53.8 | 19 it | 6523.7 |
| STaR-Acc | 73.2 | 18 it | 639.8 | 74.6 | 19 it | 1745.3 | 94.2 | 28 it | 9663.0 |
| STaR-Acc-Full | 71.8 | 5 it | **135.8** | 76.0 | 10 it | 1158.3 | 94.2 | 15 it | 4465.4 |
| STaR-Acc-Full-K | 71.4 | 3 it | 302.2 | 73.0 | 4 it | 1760.9 | 80.0 | 6 it | 6382.3 |
| ReST$^{EM}$ | 70.8 | 4 it | 637.1 | 72.8 | 2 it | 1548.4 | 53.4 | 5 it | 10498.3 |
| B-STaR | 67.8 | 2 it | 222.8 | 68.4 | 2 it | 800.9 | 52.8 | 4 it | 3937.3 |
| AdaSTaR (ours) | **73.8** | 10 it | 174.4 (↓ 72.7%) | **78.0** | 20 it | **779.3** (↓ 32.7%) | **95.6** | 23 it | **3610.0** (↓ 19.2%) |
| Evaluation | ANLI | | | GSM8K | | | SVAMP | | |
| Metric | Acc. (↑) | $t$ | PFLOPs (↓) | Acc. (↑) | $t$ | PFLOPs (↓) | Acc. (↑) | $t$ | PFLOPs (↓) |
| SFT | 64.2 | 4 e | 262.9 | 61.0 | 2.5 e | 177.3 | 57.0 | 5.5 e | 21.7 |
| SFT + 8-CoT | 65.2 | 5 e | 328.7 | 68.0 | 1 e | 70.9 | 61.5 | 7.5 e | 29.6 |
| SFT + 5-SC | 49.2 | 2 e | 131.5 | 67.2 | 2.5 e | 177.3 | 61.5 | 5.5 e | 21.7 |
| STaR | 61.0 | 23 it | 4195.3 | 76.0 | 4 it | 409.2 | 71.0 | 20 it | 373.8 |
| STaR-Full | 57.6 | 13 it | 2604.6 | 72.6 | 4 it | 684.8 | 57.5 | 37 it | 348.5 |
| STaR-Acc | 64.8 | 22 it | 3528.4 | **77.0** | 3 it | 305.2 | 71.5 | 10 it | 106.2 |
| STaR-Acc-Full | 64.6 | 5 it | **986.0** | 74.6 | 2 it | 333.0 | 74.0 | 18 it | 167.3 |
| STaR-Acc-Full-K | 58.8 | 4 it | 2528.4 | 77.0 | 2 it | 1456.5 | 75.0 | 7 it | 229.3 |
| ReST$^{EM}$ | 63.0 | 9 it | 10938.5 | 77.0 | 2 it | 2229.1 | 75.0 | 4 it | 247.8 |
| B-STaR | 59.4 | 10 it | 6373.4 | 73.6 | 3 it | 2120.2 | 72.0 | 5 it | 228.9 |
| AdaSTaR (ours) | **66.8** | 21 it | 1340.9 (↓ 62.0%) | 77.0 | 2 it | **19.3** (↓ 93.7%) | **75.5** | 9 it | **65.7** (↓ 71.3%) |

computational cost metric as memory usage remains approximately constant across methods. FLOPs are computed empirically following the method used by Kaplan et al. (2020), Sardana et al. (2024).

**Baselines.** We categorize our baselines into two groups: **(1) Vanilla SFT methods:** Regular SFT, SFT with 8-shot chain-of-thought prompting (SFT + 8-CoT; Wei et al., 2022b), and SFT with 5-sample self-consistency decoding (SFT + 5-SC; Wang et al., 2023b) with temperature 0.7.

**(2)** STaR **variants:** First, STaR (Zelikman et al., 2022), and STaR-Acc where the model is accumulated instead of being reset every iteration $t$. Most works that build on STaR choose to accumulate the model over iterations. We incorporate AdaSTaR on STaR-Acc, as STaR consistently performs empirically worse. Next, STaR-Full and STaR-Acc-Full, which is an alternative approach to eliminating the CoT sampling inefficiency described in Remark 1. In -Full, the predetermined $\beta^t$ is replaced with the total number of correct samples, i.e., $|\mathcal{D}_+^t|$. Therefore, no adaptive observation sampling scheme can be used when implementing -Full. Peng et al. (2025)'s underlying algorithm can be viewed as STaR-Acc-Full. Additionally, we include STaR-Acc-Full-K where -K denotes a larger CoT generation sample size $K$. The majority of STaR-based methods (Hosseini et al., 2024; Pang et al., 2024; Zeng et al., 2025; Lin et al., 2025) adopt -Full-K as their core strategy. In our experiments we set $K := 5$ as larger $K$ did not meaningfully improve performance, while dramatically raising compute cost. Furthermore, for -K, we omit rationalization (i.e., providing ground truth as a hint), as prior works in this setting do not employ it.

We include ReST$^{EM}$ (Singh et al., 2024), an improvement over RFT (Yuan et al., 2023) mentions the under- and over-training imbalance we discuss in § 2.2. ReST$^{EM}$ utilizes a cut-off threshold per observation $i$ to ensure training diversity. Finally, we include B-STaR (Zeng et al., 2025) with outcome verification for insight. B-STaR is the only method that builds on STaR with open-source code, allowing for faithful replication. Although Lean-STaR (Lin et al., 2025) is open-source, it is tailored to mathematical theorem proving and thus incompatible with our benchmarks.

## 4.2 Results

We first briefly discuss the baselines' performance. As organized in Tab. 1, although STaR-based approaches often outperform SFT in accuracy, they incur substantially compute costs (measured in FLOPs). Aligned with the existing literature's tendency to use model accumulation (-Acc), we see that

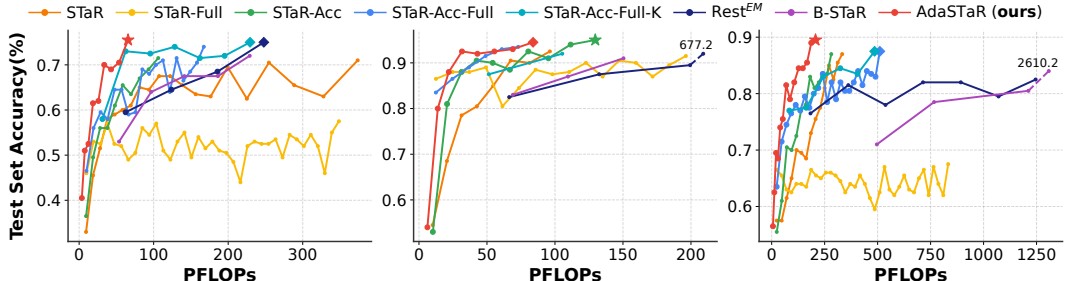

Figure 4: Visualizing the entire learning curve for SVAMP on `Llama 3.2 3B` (left), `Qwen 2.5 3B` (center), and `Gemma 7B` (right). Each method's curve is charted up to its best (early-stopped) iteration. The highest test accuracy is marked as a star, and second best as a diamond. As some methods converge only after a significant amount of PFLOPs, for legibility of shorter curves, we use dashed lines, and annotate the precise PFLOPs cost on the chart.

no model accumulation in the case of `STaR`, `STaR-Full`, and `ReST`$^{EM}$ commonly performs poorly. However, contrary to existing approaches' large $K$, we do not necessarily observe performance improving as we scale $K$. `ReST`$^{EM}$ uses $K = 11$, `STaR-Acc-Full-K` and `B-STaR` uses $K = 5$, and all remaining approaches use $K = 2$.

Comparing our `AdaSTaR` to baselines, `AdaSTaR` performs best in terms of accuracy in 6 of 6 benchmarks relative to 10 baselines, all while reducing training FLOPS by a mean of 58.6% (minimum of 19.2% to a maximum of 93.7%) relative to the strongest accuracy-achieving baseline. If there are numerous tied best baselines, we use the *lowest* PFLOPs to be conservative. Finally, for an intuitive visual understanding of our `HieMinHeap`, we provide empirical visualizations in Appendix § G.

To further evaluate generality, we test `AdaSTaR` on datasets that perform relatively weakly on `Llama 3.2 3B` using different base models and sizes. Therefore, on `Qwen 2.5 3B`, well known to be strong on mathematical reasoning, we experiment on ARC-C, GSM8K, and SVAMP. On `Gemma 7B` we experiment on ARC-C, ANLI, and SVAMP, as we observe that all methods perform significantly worse on GSM8K, relative to `Qwen 2.5 3B`. Among these five datasets (GSM8K is excluded as this is in the main text), `AdaSTaR` achieves best test accuracy 4 of 5 times, while demonstrating similar levels of training cost (FLOPs) reduction. Comprehensive results are presented in Appendix § H (`Qwen 2.5 3B`) and § I (`Gemma 7B`).

For an intuitive visualization across different base models, we visualize the entire learning curve trained on SVAMP for `Llama 3.2 3B`, `Qwen 2.5 3B`, and `Gemma 7B` in Fig. 4. Notably, across all three base models, `AdaSTaR` achieves faster gains in test accuracy under equal compute budgets. This aligns with the findings of Singh et al. (2024), which empirically demonstrate that performance gains from `STaR`-based approaches transfer well to larger-scale base models.

### 4.3 Ablation Study: Role of Diversity and Curriculum Design Choices

**Set-up.** To gain a more granular understanding of the adaptive sampling mechanism, we evaluate three ablation variants of `AdaSTaR` and analyze the standard deviation (SD) of observation training frequencies to assess whether the under- and over-training patterns observed in Fig. 3a are mitigated. The first version is `AdaSTaR` without (wo.) `AdaC`, which is synonymous to `AdaD`. Secondly, `AdaSTaR` wo. $w_i$, which changes the `HieMinHeap` to a regular `MinHeap`, only considering the last sampled iteration $\tilde{t}_i$ for priority. Finally, we experiment with a priority-flipped version (`AdaSTaR-PF`), which prioritizes $w_i$ first and $\tilde{t}_i$ second.

**Results.** We provide empirical results in Tab. 2, including `STaR-Acc` as `AdaSTaR` is mounted on top of `STaR-Acc`. Aligned with the described theory in § 3, `AdaD` (`AdaSTaR` wo. `AdaC`) most effectively reduces under- and over-training on average (↓ SD). However, contrary to the intuitive expectation that increased diversity (↓ SD) would improve test accuracy, we observe a sharp decline. We see that including `AdaC` solves this problem effectively while simultaneously maintaining high levels of trained observation diversity (↓ SD).

Table 2: Ablation empirical results with Accuracy (↑), and Standard Deviation (SD). SD of observations' trained frequency distribution is computed from iterations 1 to 2, 1 to 10, or 1 to 20 for benchmarks that converge very quickly (GSM8K), quickly (ARC-C, SVAMP), or slowly (CQA, CLadder 1.5, ANLI), respectively. Largest Acc. and lowest SD is **bolded**, and second is underlined.

| Evaluation | ARC-C | | CQA | | CLadder 1.5 | | ANLI | | GSM8K | | SVAMP | | Average | |
|---|---|---|---|---|---|---|---|---|---|---|---|---|---|---|
| Metric | Acc. | SD | Acc. | SD | Acc. | SD | Acc. | SD | Acc. | SD | Acc. | SD | Acc. | SD |
| STaR-Acc | 73.2 | 1.50 | 74.6 | 1.11 | 94.2 | 1.36 | 64.8 | 1.07 | **77.0** | 0.47 | 71.5 | 4.78 | 75.9 | 1.72 |
| AdaSTaR wo. AdaC | 72.0 | **1.14** | 74.4 | **0.90** | 52.4 | 1.13 | 65.8 | **0.88** | 75.4 | **0.00** | 70.0 | 4.61 | 68.3 | **1.44** |
| AdaSTaR wo. $w_i$ | 73.6 | 1.39 | 74.6 | 0.92 | 93.4 | 1.13 | 64.0 | 0.92 | 76.8 | 0.33 | 73.0 | 5.19 | 75.9 | 1.65 |
| AdaSTaR-PF | 72.4 | 1.82 | 74.8 | 1.00 | 67.8 | 1.24 | 64.4 | 0.98 | **77.0** | 0.32 | 72.0 | 5.02 | 71.4 | 1.73 |
| AdaSTaR (**ours**) | **73.8** | 1.26 | **78.0** | 0.99 | **95.6** | **1.12** | **66.8** | 1.04 | **77.0** | 0.32 | **75.5** | **3.98** | **77.8** | 1.45 |

AdaSTaR wo. $w_i$ does indeed, on average, reduce SD, but fails to meaningfully improve test accuracy. Therefore, we can conclude that leveraging $w_i$ to induce sampling more challenging observations within tied $\tilde{t}_i$ is a salient part of AdaSTaR. We can decompose the rise in training diversity by quantifying the fall in SD throughout STaR-Acc → AdaSTaR wo. $w_i$ → AdaSTaR: $1.72 \to 1.65 \to 1.45$. AdaSTaR-PF fails to reduce SD, as it aggressively samples challenging observations ($\downarrow w_i$), resulting in frequent resampling of difficult examples. It also results in worsened test accuracy, likely due to poorer CoT quality (see § 3.2).

## 5    Discussion and Additional Empirical Takeaways

We first discuss salient aspects of our adaptive sampling mechanism in AdaSTaR (**1, 2**), then present additional empirical insights drawn from extensive experiments with datasets and baselines under the STaR framework (**3, 4**).

(1) **Near Zero Compute Cost Statistics.**    Notably, AdaSTaR's observation sampling algorithm adapts based on three statistics: $\tilde{t}_i$, $w_i$, and $\alpha$, which costs virtually no overhead run-time to compute. While the HieMinHeap does incur some run-time compute, it is negligibly minor. Our empirical tests indicate that run-time overhead is near zero relative to the (inference) sampling and training stage. The same can be said for the minimal memory footprint. Therefore, AdaSTaR is a lightweight extension that measures and leverages statistics extractable within the existing STaR system.

(2) **Balancing Diversity and Difficulty through Adaptive Sampling.**    Our key finding is that promoting observation diversity ($\downarrow$ SD) while regularizing for model strength consistently improves performance and reduces training compute cost (Tab. 1, 5, 6). Our ablation study (Tab. 2) confirms that only encouraging inference diversity without a difficulty measure (AdaSTaR wo. $w_i$) does not lead to performance improvement. However, we also observe that failing to regularize for difficulty when the model is weaker (AdaSTaR wo. AdaC) leads to even worse outcomes. Thus, adaptively sampling more challenging observations becomes increasingly effective as model strength improves.

(3) **Model Accumulation is Generally Better.**    As seen in Tab. 1 (and also supported by Tab. 5, 6), using model accumulation (-Acc) consistently leads to improved performance. Across all experiments in the main text and Appendix, transitioning from STaR to STaR-Acc, and from STaR-Full to STaR-Acc-Full, leads to average accuracy improvements: $73.6\% \to 79.0\%$ and $67.8\% \to 78.8\%$, respectively, along with a corresponding average reduction in FLOPs of 16.4 % and 37.9%. This result is particularly noteworthy given that recent literature is divided on the use of -Acc, with some adopting it (Pang et al., 2024; Zeng et al., 2025; Lin et al., 2025; Peng et al., 2025), while others omit it (Zelikman et al., 2022; Hosseini et al., 2024; Singh et al., 2024).

(4) **Cold Starting with STaR Does Not Always Work.**    We empirically find that the viability of self-improvement via STaR depends on the difficulty of the task relative to the strength of the base model. Therefore, as discussed in § 4.1 and Appendix F, while STaR-based approaches fail to self-improve on Llama 3.2 3B, self-improvement can be realized on the better pre-trained Qwen 2.5 3B. This potentially explains why Peng et al. (2025) uses an instruction-tuned base model instead of cold starting from a pre-trained-only model. Similarly, Hosseini et al. (2024) and Zeng et al. (2025) includes an SFT stage prior to the self-improvement stage. Aligned with recent large reasoning

model training (DeepSeek-AI, 2025; Kimi Team, 2025; Liu et al., 2025b), the key takeaway is that a `STaR`-based algorithm is part of a larger training pipeline. Precisely which stage within the training pipeline it should be integrated into is an open problem.

# 6    Limitation and Future Work

We discuss relevant limitations, to the best of our knowledge, and avenues for future research. First, a natural direction for future work is to explore combinations of `AdaSTaR` with other advanced `STaR`-based methods. For instance, investigating the performance of a combined `AdaSTaR` and an inference-time verifier, such as that in `V-STaR` (Hosseini et al., 2024), presents a promising research avenue. Such explorations are beyond the scope of the current study. Second, while our experiments demonstrate `AdaSTaR`'s efficacy, a larger computational budget would have permitted evaluation on even larger-scale models. Nevertheless, our empirical study provides robust evidence of `AdaSTaR`'s effectiveness across three distinct models: `Llama 3.2 3B`, `Qwen 2.5 3B`, and `Gemma 7B`. Moreover, existing work (Singh et al., 2024) suggests that gains from `STaR`-based training on smaller models often amplify on larger scales, implying our findings may well extend or even strengthen with increased model size. Furthermore, the model sizes used in our study (up to 7B parameters) are comparable to those in related `STaR` literature (Zelikman et al., 2022, 2024; Zeng et al., 2025) that uses 6 to 7B base models. Third, similar to other adaptive methods such as `Adam` (Kingma and Ba, 2015) and `AdaGrad` (Duchi et al., 2011), `AdaSTaR` introduces a new hyperparameter $f(\alpha) := \alpha^2$. A more granular tuning is deferred to future work. It is anticipated that such tuning could lead to further enhancements in `AdaSTaR`'s performance and efficiency. Finally, building upon our discussion (§ 5), a salient direction for future work is to investigate the optimal integration of various `STaR-based` methods within the end-to-end training pipeline incorporating RL-style long CoT generation. This investigation is particularly pertinent given the current divergence in methodologies: the `STaR` stage is implemented either prior to RL (Kimi Team, 2025; Liu et al., 2025b) or subsequent to it (DeepSeek-AI, 2025). Furthermore, a critical open question is whether, and to what extent, enhancements achieved during the `STaR` phase directly propagate to performance gains in the subsequent RL stage.

Lastly, we discuss broader impact in Appendix § J.

## Acknowledgments and Disclosure of Funding

This work was improved by collaborating with researchers at LG AI Research. J. Lee and S.-Y. Yun were supported by the Institute of Information & Communications Technology Planning & Evaluation (IITP) grant funded by the Korea government(MSIT) (No. RS-2022-II220311, Development of Goal-Oriented Reinforcement Learning Techniques for Contact-Rich Robotic Manipulation of Everyday Objects, No. RS-2024-00457882, AI Research Hub Project, and No. RS-2019-II190075, Artificial Intelligence Graduate School Program (KAIST)).

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

# A   Total Training Time Comparison

Fig. 5 is conducted using full fine-tuning of `Llama 3.2 3B` (Llama Team, 2024). The training run-time is set to the early-stopped epoch (iteration) (Caruana et al., 2000).

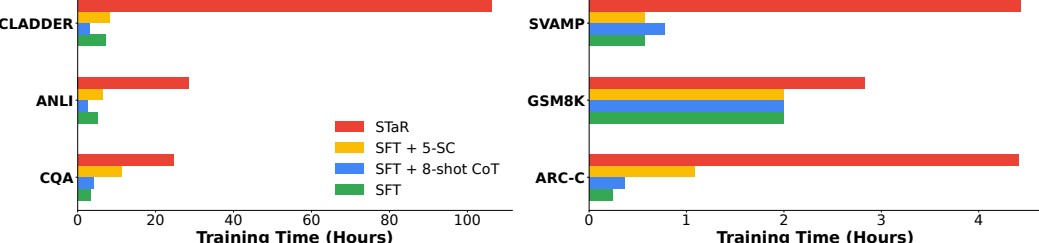

Figure 5: Total training run-time on 4×RTX 3090 24G, across three common reasoning datasets CLadder 1.5, ANLI, CQA, SVAMP, GSM8K, and ARC-C. STaR, SFT, and SFT 8-shot Chain-of-Thought is evaluated under zero-shot greedy decoding. Training times across SFT 5-sample Self-Consistency, SFT 8-shot Chain-of-Thought, and SFT differ as their best early-stop epoch differs.

# B Observation Distribution Visualizations Across All Datasets

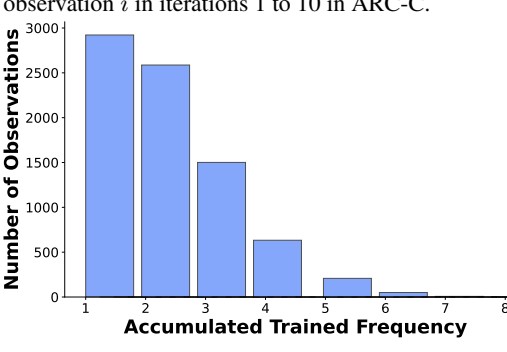

(a) Distribution of frequency trained of each observation $i$ in iterations 1 to 10 in ARC-C.

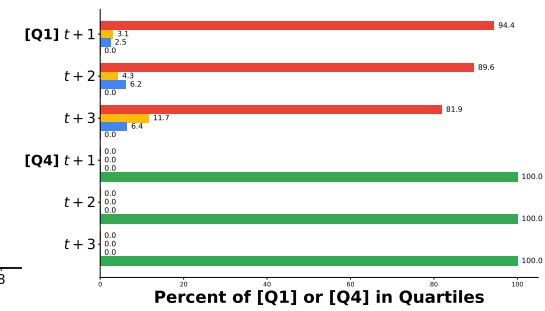

(b) Initial Quartile 1 [Q1] and 4 [Q4] are computed based on iterations 1 to 10.

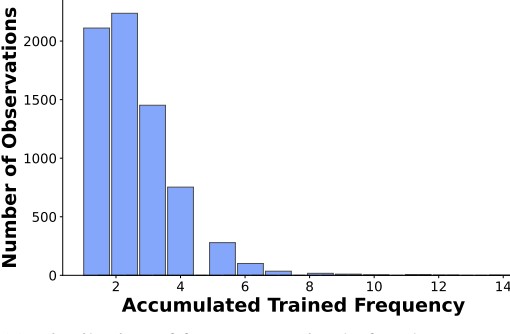

(c) Distribution of frequency trained of each observation $i$ in iterations 1 to 10 in CQA.

(d) Initial Quartile 1 [Q1] and 4 [Q4] are computed based on iterations 1 to 10.

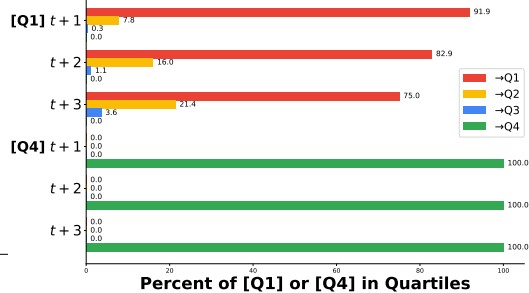

(e) Distribution of frequency trained of each observation $i$ in iterations 1 to 10 in CLadder 1.5.

(f) Initial Quartile 1 [Q1] and 4 [Q4] are computed based on iterations 1 to 10.

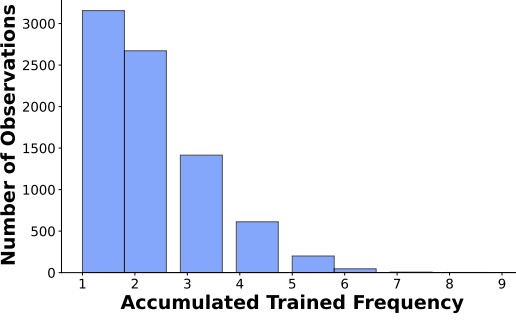

(g) Distribution of frequency trained of each observation $i$ in iterations 1 to 10 in ANLI.

(h) Initial Quartile 1 [Q1] and 4 [Q4] are computed based on iterations 1 to 10.

Figure 6: ARC-C (a, b), CQA (c, d), CLadder 1.5 (e, f), and ANLI (g, h) datasets illustrate persistent relative under- and over-training across observations.

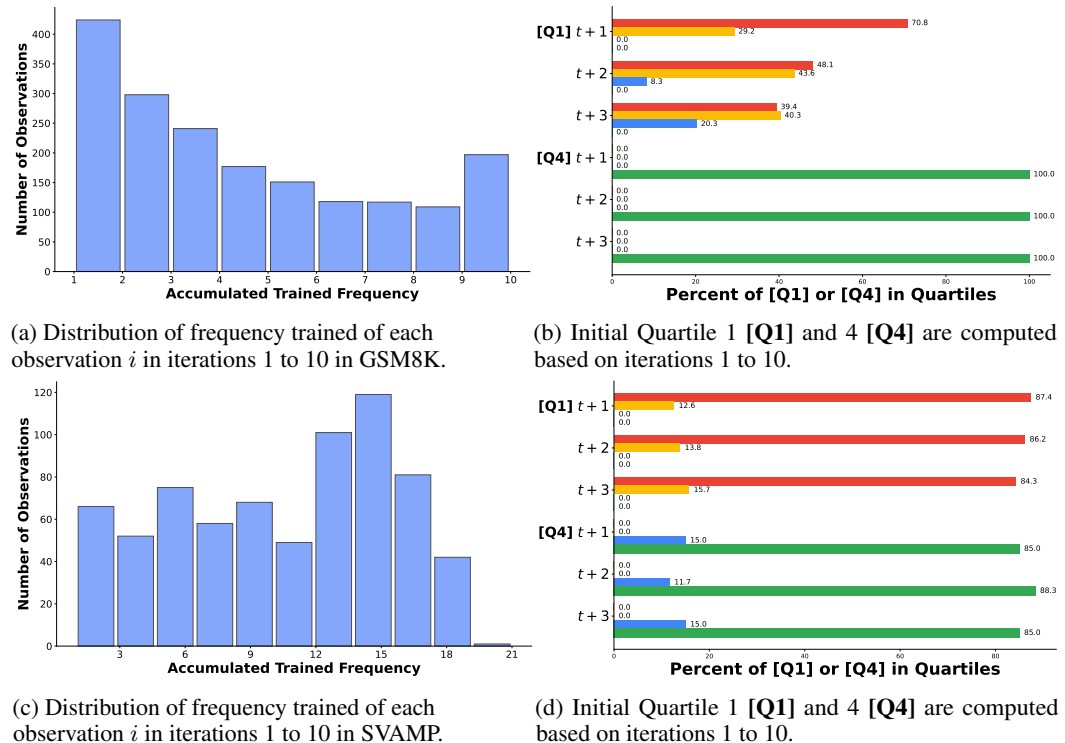

(a) Distribution of frequency trained of each observation $i$ in iterations 1 to 10 in GSM8K.

(b) Initial Quartile 1 [Q1] and 4 [Q4] are computed based on iterations 1 to 10.

(c) Distribution of frequency trained of each observation $i$ in iterations 1 to 10 in SVAMP.

(d) Initial Quartile 1 [Q1] and 4 [Q4] are computed based on iterations 1 to 10.

Figure 7: GSM8K (a, b) and SVAMP (c, d) datasets also show consistent patterns of over- and under-training across sample quartiles.

## C  AdaD Induces Poor Quality CoT

We use the following prompt.

---
**Prompt for GPT 4o Annotator**

System: You are an expert QA/CoT reviewer. For each entry, check whether the chain-of-thought (CoT) logically supports the answer. Respond ONLY with an array of the indices (0-based) of entries whose CoT is WRONG. Please output like: [1, 6], and explain why

User: [ { question : ... , chain_of_thought: ... , answer: ... }, ...
{ question : ... , chain_of_thought: ... , answer: ... }]

---

We now provide a qualitative example. Consider this example question.

---
**Question ($x_i$) Example**

*There are 6 houses on a block. If a mailman has to give 24 pieces of junk mail to each block, how many pieces of junk mail should he give in each house?*

---

Using this question, we examine the CoTs generated by AdaD (red container) and STaR-Acc (green container).



**Incorrect CoT (AdaD)**

**Given CoT:**

> 6 blocks, and 24 pieces of junk mail for each block.
>
> So, 24 pieces for 6 blocks.
>
> Then calculates: $\dfrac{24}{6} = 4$ pieces per house.

**Error Explanation:**

- Misinterpreted "6 houses on a block" as "6 blocks."
- Incorrectly calculated division using blocks instead of houses.





**Correct CoT (STaR-Acc)**

**Given CoT:**

> 6 houses per block, total of 24 pieces per block.
>
> Thus, each house gets: $\dfrac{24}{6} = 4$ pieces.

**Correct Steps:**

- Correctly identified number of houses per block.
- Correctly distributed mail equally to each house.



# D  Excessive CoT Sampling Inefficiency

## D.1  Problem: Excessive Sampling, then Filtering

A salient observation we make is that an unnecessarily large amount of CoT samples are unused in training. Remember that, `STaR` inferences the entire dataset $\mathcal{D}$[3], $\{\langle x_1, \hat{c}_1, \hat{y}_1 \rangle, \cdots, \langle x_N, \hat{c}_N, \hat{y}_N \rangle\}$, then filters down to correct samples $\mathcal{D}_+^t := \{\langle x_i, \hat{c}_i, \hat{y}_i \rangle | \mathbb{I}(y_i = \hat{y}_i)\}$. We denote the size $|\mathcal{D}_+^t| = M^t$. Next, it throws away or re-uses parts of $\mathcal{D}_+^t$[4] to fit the predetermined per iteration batch size $\beta^t$. As mentioned in § 2.1, $\beta^t = \sigma^t \times \beta$, where $\sigma^t$ is the number of gradient update steps per iteration and $\beta$ is the batch size for each gradient update step.

In the case that $M^t > \beta^t$, some $\langle x_i, \hat{c}_i, \hat{y}_i \rangle$, are discarded. Such discarded samples can not be cached and use in the next iteration $t + 1$ because the fundamental idea of iterations is that an improved model $\pi_\theta^{t+1}$ is used to generate new samples. The compute and memory wastage, especially in earlier iterations, is significant. For a concrete understanding, we visualize this sampling inefficiency empirically across the datasets in Fig. 8.

## D.2  Existing Solution

However, as mentioned in § 2.1, all methods that resolve this excessive sampling ($M^t - \beta^t$) problem of `STaR` (Hosseini et al., 2024; Pang et al., 2024; Zeng et al., 2025; Lin et al., 2025; Peng et al., 2025) simply removes this pre-determined $\beta^t$, and instead set $\beta^t$ to $|\mathcal{D}_+^t|$. This approach can be viewed as bringing the blue curve up to the red curve; i.e., $\beta^t \leftarrow |\mathcal{D}_+^t|$. We name this approach as `-Full` in our experiments (§ 4). `B-STaR` (Zeng et al., 2025) also embodies this approach. `ReST`$^{EM}$ (Singh et al., 2024) does not resolve this problem as they keep the filtering mechanism, as it is useful in their algorithm.

---

[3]This can be known from line 3 in Alg. 1 in the `STaR` paper. This is also the case in their open-source code.
[4]Excluding some rare edge scenarios $M^t = \beta^t$

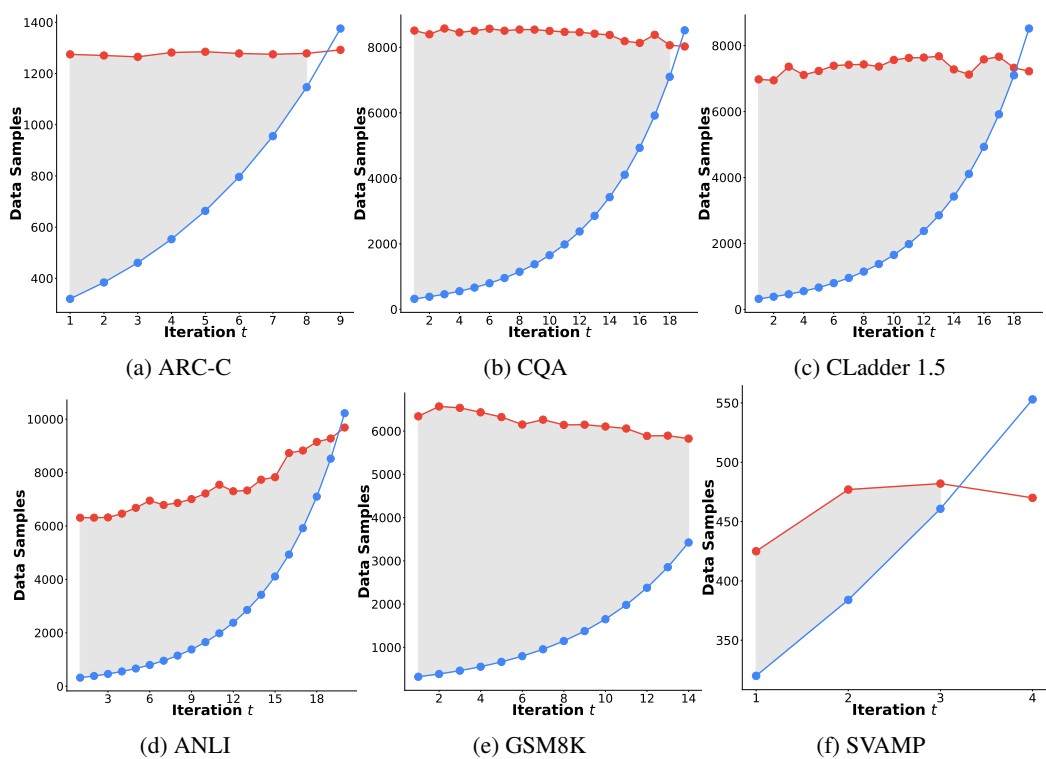

Figure 8: Visualizing the CoT sampling inefficiencies in STaR across numerous datasets. Initial $\beta^{t=1} = 40$ as presented in the original implementation. $\beta^t$ rises over time as we follow the $+20\%$ of gradient update steps $\sigma$ every iteration in the original implementation. That is, $\beta^{t+1} := 1.2(\beta^t)$. If $M^t > \beta^t$, there is an inference sampling inefficiency as $M^t - \beta^t$ data samples are not used.

### D.3 Our Approach

Alternatively, as we aim to reach peak performance as computationally efficiently as possible, we keep STaR's original $\beta^t$ curve, and instead, sample CoTs $\langle x_i, \hat{c}_i, \hat{y}_i \rangle \leftarrow \pi_\theta^t(e, x_i)$ up till $|\mathcal{D}_+^t| = \beta^t$ is filled. This approach can be viewed as bringing the red curve down to the blue curve; i.e., $|\mathcal{D}_+^t| \leftarrow \beta^t$.

## E   Further Details on Experimental Configuration and Setting

**Common Configuration.**   We primarily conduct our experiments on numerous nodes with $8 \times$RTX 3090 24G, with equivalent hardware specifications across nodes. For a few compute heavy experiments we use nodes with $8 \times$A100 40G. All training is done on the same arbitrary seed value of 10. This value has never been changed. Hyperparameters are organized in Tab. 3.

| Parameters | ARC-C | CQA | CLadder 1.5 | ANLI | GSM8K | SVAMP |
|---|---|---|---|---|---|---|
| Batch size | 8 | 8 | 8 | 8 | 8 | 8 |
| Learning rate | $10^{-5}$ | $10^{-5}$ | $10^{-5}$ | $10^{-5}$ | $10^{-5}$ | $10^{-5}$ |
| Weight decay | 0.01 | 0.01 | 0.01 | 0.01 | 0.01 | 0.01 |
| Warm up steps | 100 | 100 | 100 | 100 | 100 | 100 |
| Optimizer | Adam | Adam | Adam | Adam | Adam | Adam |
| Model precision | bf16 | bf16 | bf16 | bf16 | bf16 | bf16 |
| Samples for self consistency | 5 | 5 | 5 | 5 | 5 | 5 |
| Inference decoding temperature | 1.0 | 1.0 | 1.0 | 1.0 | 1.0 | 1.0 |
| Evaluation decoding temperature | 0 | 0 | 0 | 0 | 0 | 0 |
| Rationalization (default) | True | True | True | True | False | False |

Table 3: Hyperparameters across datasets.

**Dataset Configuration.**    For ARC-C, we combined the train and validation dataset for training. The ANLI dataset is comprised of R1, R2, and R3 versions. For our experiment, we used R1, and random sampled (without replacement) 10,000 samples for efficient evaluation. In GSM8K, high quality ground truth $c$ is already available in the SFT dataset. To compare whether STaR is able to improve on the SFT case where high quality ground truth $c$ is unavailable, we do not include the $c$ in the SFT dataset. That is, we only train on $\langle x, y \rangle$ as all STaR-like approaches are not given access to $c$. Dataset and evaluation sizes are provided in Tab. 4.

| Dataset | Train set | Test set |
|---|---|---|
| ARC-C | 1,418 | 1,172 |
| CQA | 9,741 | 1,140 |
| CLadder 1.5 | 8,089 | 2,023 |
| ANLI (R1) | 10,000 | 1,000 |
| GSM8K | 7,473 | 1,319 |
| SVAMP | 800 | 300 |

Table 4: Train and test set sizes for each dataset

ReST$^{EM}$ **Configuration.**    We follow the original implementation's ReST$^{EM}$ configuration (Singh et al., 2024) as close as possible. The only change we make is reducing $K := 32$ and cut-off threshold value of 10 to $K := 11$ and cut-off threshold value to 3. This is done as larger $K$ and cut-off threshold values resulted in worsened performance with dramatic rise in compute cost. We kept the ratio of $K$ to cut-off threshold as close to the paper's implementation.

For instance, when sampled $K = 11$, an easy observation $i$ may result in 8 correct samples, while more challenging ones may result in 2. In this case, if the threshold is set to 3, the observation with 8 correct $\langle x_i, \hat{c}_i, \hat{y}_i \rangle$ will be reduced to a maximum of 3, shrinking the imbalance from 8:2 to 3:2.

B-STaR **Configuration.**    We follow the original implementation's B-STaR configuration presented in their paper (Zeng et al., 2025) as close as possible. For any implementation that is not explicitly specificed in the paper, we use their official open-source implementation. We set the range of temperature search space as $[0.4, 1.1]$ in increments of 0.1 as in the paper. We set $K := 5$ as in the paper. We set their balancing hyperparameter $n^\star := 6$ as in the paper. The only change we make is their training queries ($M$) per iteration. We first experimented by setting $M := 2627$ as they did for their experiments that did not include a SFT stage, pre-STaR training. However, this resulted in poor performance. In response, we set $M$ to the entire original dataset size, which helped performance.

## F  `Llama 3.2 3B` **Fails to Self-Improve on GSM8K**

STaR-based methods fail to self-improve on GSM8K using `Llama 3.2 3B` as the base model (Fig. 9). Therefore, we use `Qwen 2.5 3B` instead in the main text.

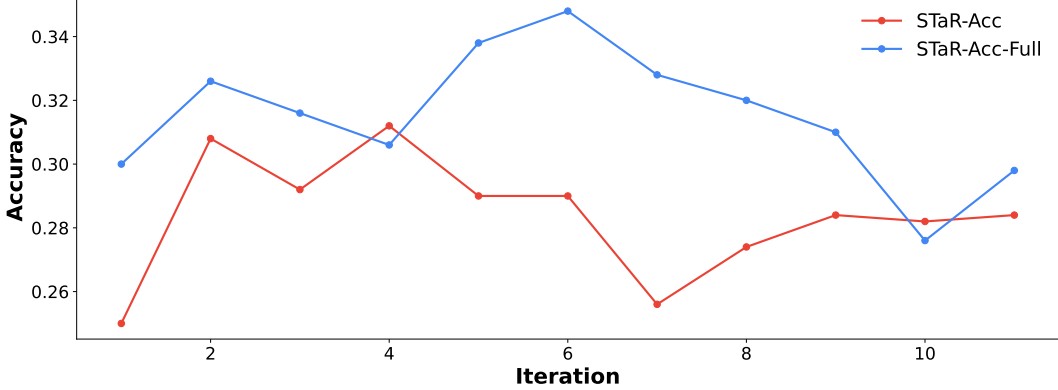

Figure 9: Visualizing the learning curve for `STaR-Acc` and `STaR-Acc-Full` for GSM8K using `Llama 3.2 3B` as the base model.

# G  Visualizing Empirical Heaps

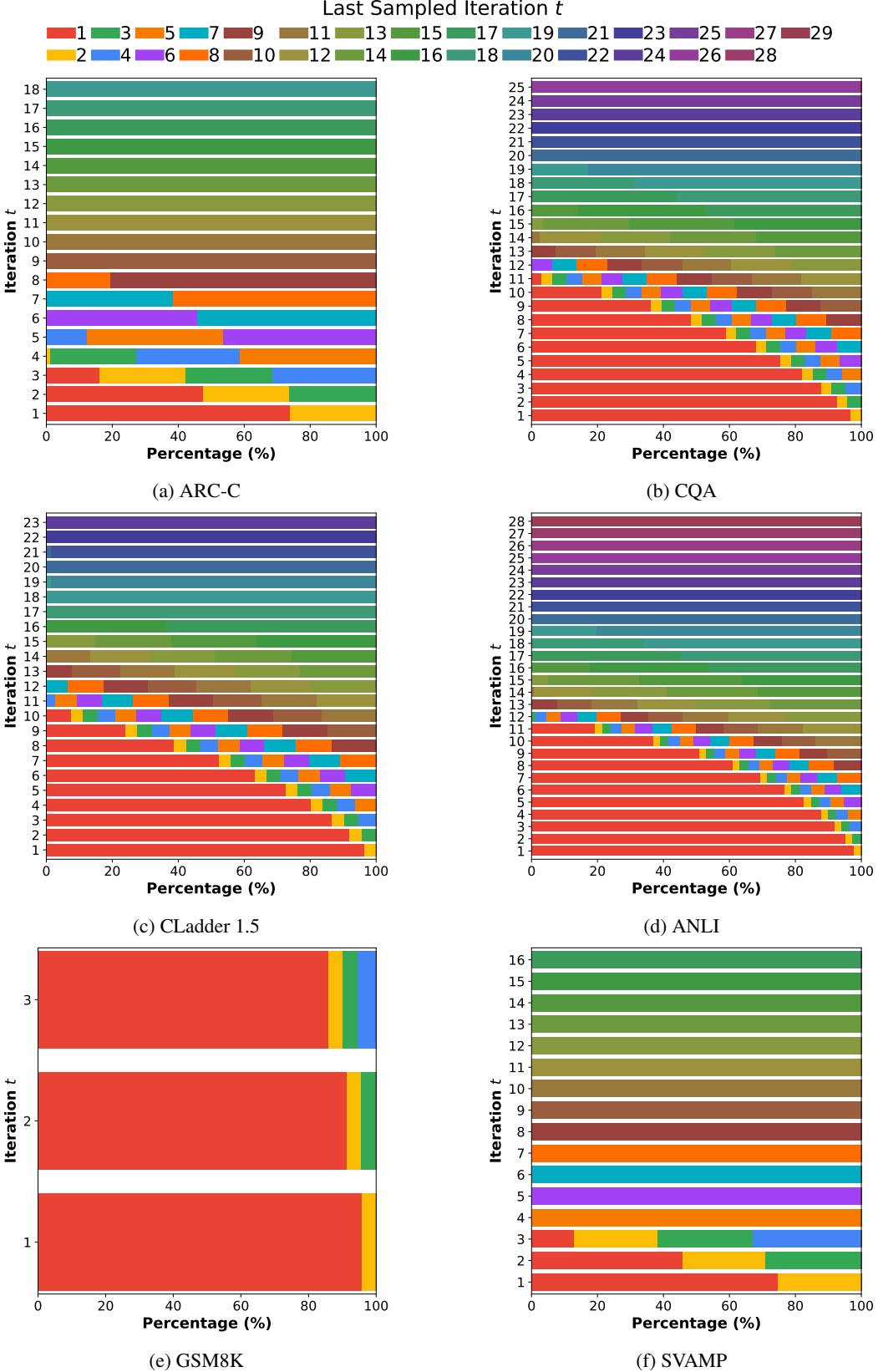

(a) ARC-C

(b) CQA

(c) CLadder 1.5

(d) ANLI

(e) GSM8K

(f) SVAMP

# H   `Qwen 2.5 3B` Base Model Empirical Results

Refer to Tab. 5 for empirical results using `Qwen 2.5 3B` as the base model. Experiment settings are equivalent to the main experiments. GSM8K in Tab. 5 is equivalent to that of Tab. 1, as the main text's GSM8K is `Qwen 2.5 3B` based. We describe why Tab. 1 is `Qwen 2.5 3B` based in § 4.1.

Table 5: `Qwen 2.5 3B` empirical results where Test Set Accuracy (%, ↑) is reported under zero-shot greedy decoding, excluding the 5-SC evaluation. Total training costs are reported in Peta FLOPs (↓). Best Acc. and PFLOPs is **bolded**, and second best is underlined in each section (excluding SFT). In (red) we quantify percent PFLOPs reduction against the highest accuracy baseline.

| Evaluation | ARC-C | | | GSM8K | | | SVAMP | | |
|---|---|---|---|---|---|---|---|---|---|
| Metric | Acc. (↑) | $t$ | PFLOPs (↓) | Acc. (↑) | $t$ | PFLOPs (↓) | Acc. (↑) | $t$ | PFLOPs (↓) |
| SFT | 33.8 | 6.5 e | 43.9 | 61.0 | 2.5 e | 177.3 | 68.5 | 0.5 e | 1.89 |
| SFT + 8-CoT | 75.2 | 7.5 e | 50.6 | 68.0 | 1 e | 70.9 | 86.5 | 4.0 e | 15.2 |
| SFT + 5-SC | 67.4 | 6.0 e | 40.5 | 67.2 | 2.5 e | 177.3 | 73.5 | 0.5 e | 1.89 |
| STaR | 80.4 | 20 it | 825.9 | 76.0 | 4 it | 409.2 | 92.5 | 8 it | 96.2 |
| STaR-Full | 83.2 | 22 it | 606.2 | 72.6 | 4 it | 684.8 | 91.5 | 16 it | 196.2 |
| STaR-Acc | 84.4 | 11 it | 264.1 | **77.0** | 3 it | 305.2 | **95.0** | 10 it | 129.6 |
| STaR-Acc-Full | 84.6 | 4 it | **110.8** | 74.6 | 2 it | 333.0 | 93.5 | 6 it | **73.0** |
| STaR-Acc-Full-K | 82.2 | 2 it | 225.1 | **77.0** | 2 it | 1456.5 | 92.0 | 2 it | 105.3 |
| ReST$^{EM}$ | 81.0 | 8 it | 874.0 | **77.0** | 2 it | 2229.1 | 92.0 | 10 it | 677.2 |
| B-STaR | 83.2 | 10 it | 583.3 | 72.6 | 2 it | 1185.7 | 91.0 | 3 it | 150.4 |
| AdaSTaR (**ours**) | **85.0** | 12 it | 239.9 (↓ 0%) | **77.0** | 2 it | **19.3** (↓ 93.7%) | 94.5 | 8 it | 83.9 (↓ 35.3%) |

# I   `Gemma 7B` Base Model Empirical Results

Refer to Tab. 6 for empirical results using `Gemma 7B` as the base model. We use Low-Rank Adaptation (LoRA; Hu et al., 2022) fine-tuning set to rank = 32. All other settings are equivalent to the main experiments.

Table 6: `Gemma 7B` empirical results where Test Set Accuracy (%, ↑) is reported under zero-shot greedy decoding, excluding the 5-SC evaluation. Total training costs are reported in Peta FLOPs (↓). Best Acc. and PFLOPs is **bolded**, and second best is underlined in each section (excluding SFT). In (red) we quantify percent PFLOPs reduction against the highest accuracy baseline.

| Evaluation | ARC-C | | | ANLI | | | SVAMP | | |
|---|---|---|---|---|---|---|---|---|---|
| Metric | Acc. (↑) | $t$ | PFLOPs (↓) | Acc. (↑) | $t$ | PFLOPs (↓) | Acc. (↑) | $t$ | PFLOPs (↓) |
| SFT | 49.2 | 0.5 e | 0.01 | 66.0 | 5 e | 1.1 | 61.0 | 3 e | 0.04 |
| SFT + 8-CoT | 76.6 | 5.5 e | 0.13 | 53.0 | 5.5 e | 1.2 | 82.0 | 4 e | 0.05 |
| SFT + 5-SC | 66.0 | 0.5 e | 0.01 | 67.6 | 7 e | 1.6 | 61.0 | 3 e | 0.04 |
| STaR | 82.0 | 17 it | 530.6 | 62.4 | 28 it | 13105.8 | 87.0 | 15 it | 332.8 |
| STaR-Full | 76.2 | 3 it | **93.5** | 43.8 | 12 it | 5536.6 | 67.5 | 36 it | 832.9 |
| STaR-Acc | **85.4** | 13 it | 383.0 | 62.0 | 20 it | 9298.8 | 87.0 | 13 it | 281.3 |
| STaR-Acc-Full | 84.6 | 20 it | 533.0 | 61.8 | 8 it | 3751.0 | 87.5 | 24 it | 510.6 |
| STaR-Acc-Full-K | 85.0 | 12 it | 1334.4 | 65.0 | 12 it | 20743.3 | 87.5 | 6 it | 485.7 |
| ReST$^{EM}$ | 81.6 | 5 it | 1221.0 | 62.2 | 17 it | 43060.8 | 82.5 | 7 it | 1245.6 |
| B-STaR | 84.8 | 15 it | 1936.0 | 63.8 | 22 it | 27315.1 | 84.0 | 7 it | 2610.2 |
| AdaSTaR (**ours**) | **85.4** | 14 it | 321.0 (↓ 16.2%) | **65.2** | 20 it | **3055.4** (↓ 85.3%) | **89.5** | 14 it | **207.0** (↓ 57.4%) |

# J   Broader Impact

The development of `AdaSTaR` presents notable positive societal benefits stemming from its ability to achieve strong performance with significantly reduced PFLOPs.

- **Environmental Sustainability:** By lowering the computational requirements (FLOPs) for training effective models, `AdaSTaR` contributes to more environmentally sustainable AI practices. This reduction directly translates to lower energy consumption and a diminished carbon footprint associated with model development and deployment.

- **Economic Value and Accessibility:** The substantial computational savings unlock economic advantages. These include reduced operational costs for training and inference, making advanced AI technologies more accessible to a broader spectrum of users. Academic institutions, startups, and researchers with limited computational budgets can benefit, potentially accelerating innovation and democratizing access to state-of-the-art model development.

- **Accelerated Research and Development:** Efficiency gains can shorten model development cycles, allowing for faster iteration and exploration of new architectures and applications.

