# OpenReview forum: "AdaSTaR: Adaptive Data Sampling for Training Self-Taught Reasoners"
_NeurIPS.cc/2025/Conference — NeurIPS 2025 poster_

### Official Review · Reviewer_5Fqt · 2025-06-24

**Clarity:** 2
**Significance:** 2
**Originality:** 2
**Rating:** 3
**Confidence:** 3

**Summary:**

The paper introduces AdaSTaR, an enhanced version of the self‑taught reasoners training loop that replaces random data sampling with two adaptive strategies: one that up‑weights examples the model has rarely seen to maintain coverage, and another that shifts the mix of easy and hard problems to match the model’s improving skill. This balanced, curriculum‑like sampling lets the model focus on challenging cases without wasting compute on items it already solves. Tested on six reasoning benchmarks, AdaSTaR attains the best accuracy in every setting  and the gains carry over to larger and different pre‑trained language models.

**Questions:**

Please refer to the Strengths And Weaknesses.

**Ethical Concerns:**

["NO or VERY MINOR ethics concerns only"]

**Final Justification:**

I am still not fully convinced by the explanation for the specific design choice of prioritizing the hardest cases within the oldest sampling time group. While I understand the motivation to address over-training of easy examples, the authors have not provided sufficient theoretical or empirical justification for why this particular ordering (oldest-first, then hardest-first) is optimal compared to alternative curriculum strategies. Additionally, this work primarily combines existing models without proposing novel theoretical contributions or formal analysis of the sampling strategy's convergence properties. Considering these limitations, I will keep my score unchanged.

**Limitations:**

Yes.

**Paper Formatting Concerns:**

Yes, the format of the reference does not follow the requirements of NeurIPS 2025 Paper Formatting Instructions.

**Quality:**

2

**Strengths And Weaknesses:**

**Strengths**

- This paper tackles a clear pain point in self-taught reasoners: the random sampling loop wastes compute on easy cases and neglects hard ones.

- The authors proposed an adaptive sampler that tracks how often each example is seen and how often the model answers it correctly, then chooses future batches to balance coverage and follow a curriculum that matches the model’s current skill.

- The experimental section is broad, covering six public reasoning datasets that span science, commonsense, causal, natural-language inference, and mathematics, plus checks on three different base models.

**Weaknesses**

- Some design choices are not well explained. For instance, the sampler first sorts items with the oldest sampling time and, within that set, prefers the hardest cases, which puts difficult problems early. It will be beneficial if the authors clarify why this order is better than the usual easy‑to‑hard order that often helps learning.

- The technical depth is not large.

- The proposed method keeps outcome verification, it may reinforces chains that reach the right answer for the wrong reason.

- The approach still requires ground-truth final answers to decide which chains are correct, so it cannot help in settings where only unlabeled questions exist, and the paper does not discuss how to relax that need.

---

> ### Author Rebuttal · Authors · 2025-07-30
>
> Thank you for your feedback and sharing your time contributing to the community. We are especially encouraged to see that you recognize:
>
> -   **Clear Takeaway**
> 	-   “tackles a clear pain point in self-taught reasoners” (`5Fqt`)
> -   **Consistently Strong Performance**
> 	-   “best accuracy in every setting … gains carry over to larger and different pre‑trained language models” (`5Fqt`)
> -   **Consistently Strong Efficiency Gains**
> 	-   “experimental section is broad … plus checks on three different base models” (`5Fqt`)
>
> We address the weaknesses **[W]** you bring-forth in detail below. Please feel free to share any further questions or concerns, as we welcome continued discussion for the broader research community.
>
> ### **[W1] Usual Easy-to-Hard**
>
> While easy-to-hard approaches are commonly used by researchers and practitioners, our approach, while related, is meaningfully different. More precisely, our main motivation for the need for easy-to-hard scheduling is different, and it is done automatically.
>
> As described in **2.2 Motivation: Need for Adaptive Data Sampling**’s **lines 122 - 140**, and visualized in **Figure 3**, we find that the random observation sampling mechanism in STaR causes a sub-optimal over-training of easy and under-training hard observations. Therefore, in our framework, we attempt to encourage sampling more challenging observations. However, up-sampling more challenging observations raises false positives, and thus, we require a curriculum-style implementation (**lines 141 - 159**).
>
> Moreover, the easy-to-hard is done wholly automatically without researcher’s manual data selection. This is an important contribution as the observation order selection is done without human labor.
>
> ###  **[W2] Technical Depth**
>
> We respectfully disagree with the assertion that our work lacks technical depth. AdaSTaR appears straightforward because we invested significant effort in making the algorithm transparent and easy to reproduce, but the underlying solution is not shallow. In fact, our contribution requires several non-trivial points that, to our knowledge, are absent from prior STaR literature:
>
> **(1)** We clearly and rigorously identify a problem in existing STaR algorithms (**Figure 3 (a), 6, 7, 8**).
>
> **(2)** We discuss, then empirically analyze why a naive solution is problematic (**Figure 3 (b)**, **Appendix C**).
>
> **(3)** We set-up a comprehensive and thorough experimental study from the perspective of performance and efficiency (**Table 1, 5, 6, Figure 4**). Moreover, we agree with Reviewer `NZVF`’s comment that the “Related work is thorough”.
>
> **(4)** We include an ablation study (**Table 2**) to thoroughly understand the components of our method, and also how the method affects the distribution of the trained observations.
>
>
>
> Furthermore, we believe that our approach’s straight-forward nature is a strength as this means that the method has been communicated clearly, and reduces replication challenges. This has been listed as a strength on the other three remaining reviews (`suRc`, `2JDr`, `NZVF`).
>
>
>
> ### **[W3] Outcome Verification**
>
> We believe that this weakness is out-of-scope as this is an orthogonal direction. The most common and *de facto* setting is where only the final answer label ($y$) is provided, making our approach most wide-reaching. Similarly, related works [1, 2, 3, 4, 5, 6] use outcome verification. Our work and others alike, achieve strong performance gains even without process verification.
>
>
> While Process Reward Models (PRMs; [7]) are an important orthogonal line of work, they require significant human annotation, stronger models, or significantly more compute to make possible.
>
>
> ### **[W4] Label-free?**
>
> We believe that this is an excellent opportunity for future work [9]; but out-of-scope for this paper. Our method and other STaR based methods [1, 2, 3, 4, 5, 6, 8] all assume the most common setting where the final answer label ($y$) is available. Training base models with labeled datasets (using e.g., AdaSTaR) is an integral and irreplaceable part of the training pipeline of LMs [10, 11].
>
> We believe that **[W3]** and **[W4]** are great opportunities for future developments, but it is challenging to cover every possible direction in a single self-contained paper.
>
>
> ### **Reference**
>
> [1] Eric Zelikman, Yuhuai Wu, Jesse Mu, and Noah Goodman. “STaR: Bootstrapping Reasoning With Reasoning”, NeurIPS 2022.
>
> [2] Xiangyu Peng, Congying Xia, Xinyi Yang, Caiming Xiong, Chien-Sheng Wu, and Chen Xing.
>
> “ReGenesis: LLMs can Grow into Reasoning Generalists via Self-Improvement”, ICLR 2025 (Oral).
>
> [3] Arian Hosseini, Xingdi Yuan, Nikolay Malkin, Aaron Courville, Alessandro Sordoni, and Rishab Agarwal. “V-STaR: Training Verifiers for Self-Taught Reasoners”, COLM 2024.
>
> [4] Richard Yuanzhe Pang, Weizhe Yuan, He He, Kyunghyun Cho, Sainbayar Sukhbaatar, and Jason E Weston. “Iterative Reasoning Preference Optimization”, NeurIPS 2024.
>
> [5] Zheng Yuan, Hongyi Yuan, Chengpeng Li, Guanting Dong, Keming Lu, Chuanqi Tan, Chang Zhou, and Jingren Zhou. “Scaling Relationship on Learning Mathematical Reasoning with Large Language Models”, Arxiv 2023.
>
> [6] Avi Singh, John D Co-Reyes, Rishabh Agarwal, Ankesh Anand, Piyush Patil, Xavier Garcia, Peter J Liu, James Harrison, Jaehoon Lee, Kelvin Xu, Aaron T Parisi, Abhishek Kumar, Alexander A Alemi, Alex Rizkowsky, Azade Nova, Ben Adlam, Bernd Bohnet, Gamaleldin Fathy Elsayed, Hanie Sedghi, Igor Mordatch, Isabelle Simpson, Izzeddin Gur, Jasper Snoek, Jeffrey Pennington, Jiri Hron, Kathleen Kenealy, Kevin Swersky, Kshiteej Mahajan, Laura A Culp, Lechao Xiao, Maxwell Bileschi, Noah Constant, Roman Novak, Rosanne Liu, Tris Warkentin, Yamini Bansal, Ethan Dyer, Behnam Neyshabur, Jascha Sohl-Dickstein, and Noah Fiedel. “Beyond Human Data: Scaling Self-Training for Problem-Solving with Language Models”, TMLR 2024.
>
> [7] Hunter Lightman, Vineet Kosaraju, Yuri Burda, Harrison Edwards, Bowen Baker, Teddy Lee, Jan Leike, John Schulman, Ilya Sutskever, Karl Cobbe. “Let's Verify Step by Step”, ICLR 2024.
>
> [8] Weihao Zeng, Yuzhen Huang, Lulu Zhao, Yijun Wang, Zifei Shan, and Junxian He. “B-STaR: Monitoring and Balancing Exploration and Exploitation in Self-Taught Reasoners”, ICLR 2025.
>
> [9] Yuxin Zuo, Kaiyan Zhang, Li Sheng, Shang Qu, Ganqu Cui, Xuekai Zhu, Haozhan Li, Yuchen Zhang, Xinwei Long, Ermo Hua, Biqing Qi, Youbang Sun, Zhiyuan Ma, Lifan Yuan, Ning Ding, Bowen Zhou. “TTRL: Test-Time Reinforcement Learning”, Arxiv 2025.
>
> [10] OLMo Team. “2 OLMo 2 Furious”, Arxiv 2025.
>
> [11] Qwen Team. “Qwen2.5 Technical Report”, Arxiv 2025.

---

> > ### Comment · Reviewer_5Fqt · 2025-08-07
> >
> > Dear Authors,
> >
> > I thank the authors for the detailed rebuttal and have no additional questions.
> >
> > Best
> >
> > Reviewer 5Fqt

---

> ### Author Response · Authors · 2025-08-05
> **Gentle Reminder**
>
> To Reviewer `5Fqt`,
>
> While we understand that everyone in our community has a lot on their plate, we would like to offer a gentle reminder, as we believe there are important discussion points that would benefit from further clarification. Please let us know if we have missed anything or misunderstood any part of your feedback.
>
> Best regards

---

### Official Review · Reviewer_NZVF · 2025-06-27

**Clarity:** 2
**Significance:** 4
**Originality:** 3
**Rating:** 5
**Confidence:** 3

**Summary:**

This paper proposes to modify STaR, an algorithm used in improving the training of many language models (LMs), so that its sampling remains both diverse and iteratively moves towards more challenging observations (curriculum learning). The authors point out several existing limitations with research involving STaR and address these issues with their proposed AdaSTaR. They test it on 6 benchmarks with a few LMs, and their proposed work achieves greater test accuracy.

**Questions:**

I quite enjoyed this paper, and am convinced by the paper's overall goal. I may consider changing my score to a Strong Accept if you can clarify some of my questions below. In particular, I am most interested in understanding the CoT exemplar notation better (my questions about lines 100 and 104) and the last question at the bottom about why use the formula chosen for capping how many updates to be made (see my questions/comments for line 234).

Line 100: The symbol "e" is used both on the left of the equation and as an index to the indexed set. The math appears wrong to me. Did you mean something like: "CoT exemplars $\mathcal{E} = \{ \langle x_{e}, c_{e}, y_{e}  \rangle \}\_{e = 1}^{E}$."? Then, shortly after, there is a reference to $c_{i}$: "However, as no ground truth $c_{i}$ is available, ..." Please confirm my understanding that the training dataset has CoT for each query, but we are not guaranteed it is the ideal CoT.

Line 102 and/or line 104/108: What is the significance of the \hat? Is it to denote that it was obtained/sampled from an LM? Edit: seems the \hat is clarified on lines 179-180, but the authors should consider moving similar language earlier to its first appearance.

Line 103: Is $k \in [K]$ saying that $k$ is an element of the equivalence class of $K$? I am unsure this was the intended meaning here, as $[ \cdot ]$ has a special but somewhat standard mathematical meaning, and $K$ was previously defined as $K \in \mathbb{N}$.

Line 104: Is $e$ referring to a specific exemplar, its associated index, or the entire set of fixed few-shot CoT exemplars? Same issue with lines 105-106. Also, for as little as RL has been mentioned in this work thus far, I assume that "If $r = 1$, it is accepted, and if $r = 0$, ..." is referring to a reward as $r$?

Algorithm 1: Why are the variables on lines 1 and 2 being defined as dictionaries when they could just be vectors like $\bar{t} = \mathbf{0}$ and $w = \mathbf{0}$? After all, they are indexed by integers, and all entries are initialized to zero (line 206). Seems unnecessarily verbose, unless you wanted the explicit size mentioned. Also, do we actually need the temporary variable defined on line 6? It is used on line 12

Line 234: Why is this formula used for determining the highest priority? Why square the model strength? Why multiply it by $m$, and why apply the floor function? Additionally, the footnote uses $f$ to denote a function, but this does not match the previous text. I am also unconvinced by this existing justification in the footnote, and cannot see the relationship between $\lfloor m \alpha^{2} \rfloor$ and why that should be used for popping the highest-priority observations. It does not seem to be very sophisticated or nuanced, but rather seems random or from ad hoc experimentation for a cutoff threshold that got decent results. Why not do something more nuanced, like dynamically determining the cutoff from the HieMinHeap?

**Ethical Concerns:**

["NO or VERY MINOR ethics concerns only"]

**Final Justification:**

Recommendation is unchanged except that a reduction of clarity from 3 (good) to 2 (fair) has been applied. I find some of the mathematics to be ambiguous and a bit sloppy, but if the authors are committed to addressing these matters, I shouldn't have further concerns.

**Limitations:**

Yes

**Paper Formatting Concerns:**

None to declare.

**Quality:**

4

**Strengths And Weaknesses:**

Strengths:
+ Impressive contribution focusing on efficiency.
+ Related work is thorough; I particularly appreciate the rigorous literature supporting the RL claim yielding long CoTs.
+ The paper is written very well overall. Most of the material is explained nicely with professional and helpful visuals.
+ The authors' proposed contribution is easy to follow, as they explained it. Their work seems to be very important to advancing LM research by improving the efficiency of STAR.
+ Experiments are well done and diverse.

Weaknesses:
- Lines 111 to 113 are poorly written/justified. The authors simply state details about $\beta^{t = 1}$, and how it follows the original implementation, but do not explain why they chose that despite them mentioning alternative STaR-based approaches remove the pre-determined $\beta^{t}$ and instead set it to the size of the correct samples.

Various suggestions for edits, some minor, others major:

Line 67: Should this be FLOPs instead of PFLOPs? Earlier in the abstract, it was the former. I believe this is a typo, but later on, in the Table 1 caption, you use Peta FLOPs.

Line 75: Change "... objective. To curate preference pairs, ..." to use a semicolon instead of the period. The sentence that follows is directly related to the previous in finishing its thought, so it might be best to make that more explicit in the writing.

Line 103 does not start the paragraph off well. Write: "Let $K \in \mathbb{N}$ denote the number of CoT traces sampled as follows..."

Line 204: No need to use shorthand notation of "w.r.t." - plenty of space to write it out completely. If used later in text, then "with respect to (w.r.t.)", but the undefined shorthand could confuse some readers.

Line 207: Consider rephrasing to: "... we utilize \citeauthor{cormen_hie_min_heap}'s Hierarchical Min Heap (\texttt{HieMinHeap})..." - it looks strange with the shortened version in-line with the longer name.

Line 229 - 230: The previous justification (lines 228 - 229) for why training accuracy is used over validation or test set accuracy is reasonable, but test set accuracy shouldn't even be an option by any means, as you are likely aware. I'd almost suggest removing its mention from the text in these lines, as using the test set accuracy to improve a model would be wrong on many different levels.

Line 261: update so it says: "... following techniques used by Kaplan et al. (2020) and Sardana et al. (2024)." instead (or similar wording)

Table 1 caption: Probably need to fully define PFLOPs before using it.

Avoid using the word "average". Change "average" on line 293 to make use of "mean". For instance, "... reducing training FLOPs by a mean of 58.6% (minimum of ...)"

---

> ### Author Rebuttal · Authors · 2025-07-30
>
> Thank you for your thoughtful feedback and valuable contribution to our research community. We are especially encouraged to see that you recognize:
>
> -   **Clear Takeaways, and Impact**
> 	-   “very important to advancing LM research” (`NZVF`)
> -   **Consistently Strong Performance**
> 	-   “Experiments are well done and diverse” (`NZVF`)
> -   **Consistently Strong Efficiency Gains**
> 	-   “Impressive contribution … on efficiency” (`NZVF`)
> -   **Straight-forward Implementation**
> 	-   “proposed contribution is easy to follow” (`NZVF`)
> -   **Clear Communication**
> 	-   “Related work is thorough” (`NZVF`)
> 	-   “written very well overall … explained nicely with professional and helpful visuals” (`NZVF`)
>
> We address the weaknesses **[W]** and questions **[C]** you bring-forth in detail below. Your contribution has helped us improve parts of the updated version of our paper. Please feel free to share any further questions or concerns, as we welcome continued discussion for the broader research community.
>
> ### **[W1] Justification on Pre-determined $\beta^t$**
>
> You raise an important point. As we address this in detail (with visualizations) in the paper we would like to direct you to **Remark 1** (**lines 214 - 216**) and **Appendix D**. To briefly summarize, we stick with the original STaR’s direction as this was logically more sample (compute) efficient.
>
> Furthermore, the alternative approach of using the “Full” correct set is embodied in numerous baselines (STaR-Full, STaR-Acc-Full, STaR-Acc-Full-K, B-STaR, and ReST$^{EM}$). We empirically validate our discussion in **Remark 1** (**lines 214 - 216**) and **Appendix D**, as seen in **lines 269 - 271**, **Table 1, 5 , 6**, and **Figure 4**. In short, our approach that chooses not to use the alternative approach (“Full”) realizes both performance and efficiency gains
>
> ### **[W2] Writing Improvements**
>
> We appreciate your thorough suggestions, and have accordingly made the following revisions:
>
> **(1)** Line 67: PFLOPs → FLOPs
>
> **(2)** Line 75:
>
> > ... objective. To curate preference pairs, …
>
> →
>
> > "... objective: to curate preference pairs, ..."
>
> **(3)** Line 103:
>
> > K \in \mathbb{N} number of CoT traces are sampled as follows…
>
>
> →
>
> > Let K \in \mathbb{N} denote the number of CoT traces sampled as follows…
>
> **(4)** Line 204: w.r.t. → with respect to
>
>
>
> **(5)** Line 207:
>
> > ... we utilize the Hierarchical Min Heap $\texttt{HieMinHeap}$ (Cormen et al., 2022) …
>
> →
>
> > ... we utilize Cormen et al. (2022)'s Hierarchical Min Heap $(\texttt{HieMinHeap}$) ...
>
> **(6)** Line 330:
>
> > ... for instance, validation or test set accuracy …
>
> →
>
> > ... for instance validation set (not used in final evaluation) accuracy …
>
> We wholly agree with your statement on not using test accuracy. To our best knowledge, it is common practice to tune models using the train + validation set, then finally evaluating on the test set [1, 2, 3]. Please let us know if this judgement seems incorrect; we will remove this sentence entirely as it is not a part of our core contribution.
>
> **(7)** Line 261:
>
> > FLOPs are computed empirically following Kaplan et al. (2020), Sardana et al. (2024).
>
> →
>
> > FLOPs are computed empirically following the method used by Kaplan et al. (2020) and Sardana et al. (2024).
>
>
> **(8)** Table 1 Caption:
>
> > Peta FLOPs (\downarrow)
>
> →
>
> > Peta FLOPs (PFLOPs, \downarrow)
>
>
> **(9)** Line 293:
>
> > 58.6% on average
>
> →
> > by a mean of 58.6%
>
> ### **[Q1] CoT Exemplar Notation**
> Thanks for pointing this out; $e$ should not have been used on both LHS and RHS. We will modify the RHS: $e$ → $\epsilon$, then correspondingly update the remaining text to follow this notation.
>
> We would like to further clarify on your questions:
>
> > Please confirm my understanding that the training dataset has CoT for each query, but we are not guaranteed it is the ideal CoT.
>
> > Is e referring to a specific exemplar, its associated index, or the entire set of fixed few-shot CoT exemplars?
>
> For each given data set $\mathcal{D}$, a few shot $e$ ($E$ in size) is commonly available as CoT exemplar [4]. Therefore, there is no query-specific $e$, rather there is a single $e$ for all queries (given the same $\mathcal{D}$). This is the *de facto* CoT setting [5, 6], used equivalently across all related works.
>
> ### **[Q2] \hat Notation**
>
> > Is it to denote that it was obtained/sampled from an LM?
>
> Yes, this is correct. Akin to **lines 179 - 180**, we would like to direct you to **line 99**, where we describe this:
>
> > To achieve this, $\pi_\theta^t$ generates $\langle \hat{c}_i, \hat{y}_i \rangle$ ...
>
> ### **[Q3] [K] Notation**
>
> We apologize for the confusion. We acknowledge that we have not explicitly defined the notation $[K]$ as $\{1, 2, \cdots, K\}$, which is quite standard in computer science (e.g., 1.1.1 Example in [7]).
>
> To avoid confusion, we will replace $[K]$ → $\{1, 2, \cdots, K\}$.
>
> ### **[Q4] r = reward?**
> Yes, we would like to direct you to **lines 101 - 102** where we mention this:
>
> > Given the supervised dataset, a rule-based verifier defines a reward signal $r := \mathbb{I}(y_i = \hat{y}_i)$, where $\mathbb{I}(\cdot)$ is the indicator function.
>
> For a more detailed discussion on the relationship between RL and STaR we would like to refer you to the following journal paper [8].
>
> ### **[Q5] Algorithm 1’s Notation**
>
> **(1)** As mentioned in **line 162**, **Algorithm 1** represents a pseudocode. Therefore, our goal was to explain the algorithm in the clearest manner possible. We chose to use dictionaries as it intuitively provides a mapping for each query $i$.
>
> Nevertheless we acknowledge that in the actual implementation a list (vector) will guarantee O(1) look-up when the index is known, thus, we are open to updating Algorithm 1 to vector-based. Please let us know if you believe that using vectors in the pseudocode is as intuitively straight-forward as the dictionary notation.
>
> **(2)**
> > Also, do we actually need the temporary variable defined on line 6? It is used on line 12
>
> We realize that this was an error on our part; this will be removed in the updated version. Our original intent was to indicate that if a query $i$ has a small win rate $w_i$, it should not be accidentally re-sampled (.peek_next) in the same iteration $t$. However, like our code implementation, this is not necessary, as peeked $i$ are not pushed back in during the same iteration $t$.
>
> ### **[Q6] Popping Mechanism for Curriculum**
>
> > Why multiply it by $m$?
>
> Recalling that $m$ is the total number of sampled observations, an implementation with no curriculum would be:
> > for $1, \cdots, m$ do
>
> However, we are attempting to regularize the algorithm through a curriculum-style mechanism where we leave some easier samples to be re-sampled next iteration (see **lines 221 - 224**). Therefore, instead of popping and pushing $m$ observations we want it to pop and push $n \leq m$ number of samples where $n \in [0, m]$. To achieve this we multiply $m$ by $f(\cdot) \in [0, 1]$. As we want $f(\cdot)$ to reflect model strength, we let $f(\alpha)$ be a function of $\alpha$.
>
> The reasoning behind the choice of $f(\alpha) := \alpha^2$ is as described in **Page 6**’s Footnote:
>
> > The choice of $f(\alpha) := \alpha^2$ is a hyperparameter. It allows more repetition of easy observations when the model is weak, and $\textit{rapidly}$ phases out this regularization effect as the model strengthens.
>
> Considering $f(\alpha) \in [0, 1], m(1 - f(\alpha))$ is the number of easy samples that are going to be mixed into the next iteration. Therefore, $\alpha^2$'s sharper curve will be a faster phase-out of this easy observation mix, as $\alpha$ increases.
>
> > why apply the floor function?
>
> This is required as we require an integer value. It is not possible to pop a non-integer real value number of observations.
>
> We further discuss the potential limitation of this design choice in **Appendix E**’s **lines 780 - 783**:
>
> > Third, similar to other adaptive methods such as Adam (Kingma and Ba, 2015) and AdaGrad (Duchi et al., 2011), AdaSTaR introduces a new hyperparameter $f(\alpha) := \alpha^2$. A more granular tuning is deferred to future work. It is anticipated that such tuning could lead to further enhancements in AdaSTaR’s performance and efficiency.
>
> > Why not do something more nuanced, like dynamically determining the cutoff from the HieMinHeap?
>
> As observed in other adaptive methods (**Appendix E**; **lines 780 - 783**), it is challenging to introduce no new hyperparameter to the system. It is unclear if there is an obvious approach to set $f(\alpha)$ directly from the HieMinHeap. We do not believe an introduction of a hyperparameter takes away from the core contribution.
>
> Considering that we have done a minimal search for $f(\alpha)$, yet achieved consistently best test accuracy and FLOPs reduction against 7 baselines and across 3 model families suggests that there is more room for improvement. We are optimistic that future empirical searches will further drive performance and efficiency gains.
>
> ### **Reference**
>
> [1] Mitchell Wortsman, et al, “Model soups: averaging weights of multiple fine-tuned models improves accuracy without increasing inference time”, ICML 2022 (Oral).
>
> [2] Sonia Meyer, et al, “A Comparison of LLM Finetuning Methods & Evaluation Metrics with Travel Chatbot Use Case”, Arxiv 2024.
>
> [3] Zhenqian Shen, et al, “Efficient Hyper-parameter Optimization with Cubic Regularization”, NeurIPS 2023.
>
> [4] Gao, Leo et al. “The Language Model Evaluation Harness”, Zenodo 2024.
>
> [5] Jason Wei, et al, “Chain-of-Thought Prompting Elicits Reasoning in Large Language Models”, NeurIPS 2022.
>
> [6] Xuezhi Wang, et al. “Self-Consistency Improves Chain of Thought Reasoning in Language Models”, ICLR 2023.
>
> [7] Stanley, Richard P. "Enumerative combinatorics volume 1 second edition." Cambridge studies in advanced mathematics (2011).
>
> [8] Avi Singh, et al. “Beyond Human Data: Scaling Self-Training for Problem-Solving with Language Models”, TMLR 2024.

---

> ### Author Response · Authors · 2025-08-05
> **Gentle Reminder**
>
> To Reviewer `NZVF`,
>
> While we understand that all of us in our community have a lot on our plates, we would like to provide a gentle reminder as we put in significant effort into responding to the questions **[Q]**, and weaknesses **[W]**. Please let us know if we missed anything, or misunderstood parts of your discussion points.
>
> Best regards

---

> ### Comment · Reviewer_NZVF · 2025-08-06
>
> [W1] '
>
> You should include a reference to the discussion in Appendix D, then, as well as clarifying this matter in the relevant lines (111 to 113). Jumping between sections shouldn't be required.
>
> [W2]
>
> Thank you for working on those suggested edits. In response to:
>
> " We wholly agree with your statement on not using test accuracy. To our best knowledge, it is common practice to tune models using the train + validation set, then finally evaluating on the test set [1, 2, 3]. Please let us know if this judgement seems incorrect; we will remove this sentence entirely as it is not a part of our core contribution."
>
> I am not saying that we do not evaluate on a test set, nor need references to this, but the writing as it exists right now (lines 229-230: "This explains our choice over, for instance, validation or test set accuracy"), suggests that test set accuracy would even be considered as an intermediate form to improve the model. Which, to the best of my knowledge, is bad practice. You should simply just state you chose training accuracy over validation accuracy for ____ reasons. My point is: it causes concern when the test dataset is potentially leaked to the steps of training a model. As the writing is now, it makes the reader believe that this was considered.
>
> [Q3] [K] Notation
>
> Yes, I understand, but it can be a bit ambiguous as there are other works out there [1] that assign different meanings (e.g., equivalence classes) to this notation, like set theory... It is actually pretty common. Simply preceding it with "set" would be enough, or introducing it like the book you referenced [7]. The reader should not assume. We also shouldn't assume whether it is zero-based, which, in your response, suggests it is not (the more common usage). Another matter to consider is $\mathbb{N}$ - it can be "natural numbers" or "non-negative integers" like in [7]. Some might include 0 in the meaning of natural numbers [2], $\mathbb{N}$, so perhaps just state this is the set of positive integers, $\mathbb{Z}^{+}$
>
> [Q5] Algorithm 1 pseudocode
>
> Yes, I think vector-based would perhaps be cleaner over the dictionary.
>
> [Q6] Popping Mechanism for Curriculum
>
> The choice of the f function: It still feels a bit arbitrary to choose this function. Is it the ideal choice? Optimal? Why not try other possible functions?
>
> [1] Zdzisław Pawlak, "Rough Sets: Theoretical Aspects of Reasoning about Data", Springer Dordrecht 1991.
>
> [2] Kenneth Rosen, "Discrete Mathematics and Its Applications", McGraw-Hill, 2011.

---

> > ### Author Response · Authors · 2025-08-07
> > **Discussion Response**
> >
> > Thank you for your thoughtful engagement with our work.
> >
> > ### **[W1] Response**
> >
> > We agree with your writing suggestion. We will include this around **lines 111 - 113** in our updated version.
> >
> > ### **[W2] Response**
> >
> > We are on the same page on this one. We will make this modification in our updated version.
> >
> > ### **[Q3] Response**
> >
> > Thanks for pointing this out. We will update our writing to make the definition explicit.
> >
> > ### **[Q5] Response**
> >
> > We will make this writing change in the updated version.
> >
> > ### **[Q6] Response**
> >
> > > Is it the ideal choice? Optimal? Why not try other possible functions?
> >
> > As mentioned in our previous response, we were unable to search through an extensive range of $f(\alpha)$ due to compute constraints. We are optimistic that more customized functions will be empirically found over time as the minimally searched $f(\alpha)$ already produces strong results. We explicitly discuss this in our existing paper.
> >
> > Nevertheless, we are continuing to work on further hyperparameter search for the remaining duration as we acquire more compute.

---

### Official Review · Reviewer_2JDr · 2025-06-28

**Clarity:** 3
**Significance:** 3
**Originality:** 3
**Rating:** 4
**Confidence:** 3

**Summary:**

AdaSTaR tweaks the classic Self-Taught Reasoner loop by picking training examples that have been neglected for a while and are still hard for the model, mixing in a few easier ones early on. The sampler is light-weight and drops straight into existing STaR code. Experiments on a broad set of reasoning tasks show better accuracy with a lot less training compute, and the gains hold across several small language  models.

**Questions:**

1. Will AdaSTaR work on training on a mixture of data from different datasets?
2. How do AdaSTaR-trained model generalize to unseen test data?
3. How do you calculate training FLOPs in detail?

**Ethical Concerns:**

["NO or VERY MINOR ethics concerns only"]

**Final Justification:**

The additional experiments strengthen the paper. But I still think AdaSTaR would need to pair with some ways of prediction on when to early-stop, in order to demonstrate its actual speed-up in a real-world setting. I decide to maintain my score.

**Limitations:**

yes

**Quality:**

3

**Strengths And Weaknesses:**

Strengths:
1. The paper clearly spots a waste pattern in vanilla STaR.
2. Implementation is simple: two counters per example and a min-heap, making it a seamless plug in to existing STaR code.
3. Tests cover diverse datasets, base models, and baselines; results are consistent and the efficiency win is compelling.

Weaknesses
1. To my best understanding, Table 1 reports accuracy when training and evaluation share the *same* dataset, leaving open whether AdaSTaR helps models generalise to new tasks; transfer experiments would strengthen the claim.
2. (Minor) The training FLOP reported in Table 1 reflect early-stopped checkpoints, not full-budget training. In scenarios where practitioners still spend the entire budget, the adaptive sampler’s faster convergence might not translate into actual cost savings, even though the speed-up is valuable in its own right.

---

> ### Author Rebuttal · Authors · 2025-07-30
>
> Thank you for your feedback and contribution to our research community. We are especially encouraged to see that you recognize:
>
> -   **Clear Takeaway**
> 	-   “clearly spots a waste pattern” (`2JDr`)
> -   **Consistently Strong Performance**
> 	-   across several small language models” (`2JDr`)
> -   **Consistently Strong Efficiency Gains**
> 	-   “a lot less training compute” (`2JDr`)
> 	-   “diverse datasets, base models, and baselines; results are consistent and the efficiency win is compelling” (`2JDr`)
> -   **Straight-forward Implementation**
> 	-   “seamless plug in” (`2JDr`)
>
> We are happy to address the weaknesses **[W]** and questions **[Q]** you bring-forth in detail below. Please feel free to share any further questions or concerns, as we welcome continued discussion for the broader research community.
>
> ### **[W1 & Q1] Transfer Generalization and Mixture of Datasets**
>
> **Transfer Generalization.** You are correct that our study is done out-of-sample (held out test set) but for the same (train, test) dataset. Task transfer evaluation, while valuable, is not the focus of existing STaR methods [2, 3, 4, 5, 6, 7, 10], as these approaches are typically designed for task-specific improvement. Additionally, transfer evaluation is particularly challenging for the smaller pre-train-only models we focus on (Llama 3.2 3B, Qwen 2.5 3B, and Gemma 7B), which have limited cross-task capabilities compared to larger models.
>
>
> **Mixture of Datasets.** Following the standard protocol of related works [2, 3, 4, 5, 6, 7, 10], we do not provide any mixture of dataset experiments. While we believe this is an excellent future direction, we believe it is fair to assume that AdaSTaR or any other STaR-based approaches can be applied to each training set. Large post-training datasets typically comprise multiple sub-datasets [11, 12, 13, 14, 15] in practice, allowing AdaSTaR to be applied to each subset independently.
>
> **Additional Experiments.** Nevertheless, to provide some experimental results during the rebuttal phase, we run simple experiments (due to computational and time constraints) inspired by [10] to show that AdaSTaR’s results remain consistent in transfer and mixture of datasets.
>
> The experiment protocol here is:
>
> **1.**  Train: GSM8K → Test: SVAMP
>
> **2.**  Train: GSM8K + SVAMP → Test: GSMK + SVAMP
>
> **3.**  Train: SVAMP + ARC-C → Test: GSM8K + CQA
>
> The base model is Qwen 2.5 3B, and the train set is randomly shuffled. All other settings remain consistent with the original paper. We choose STaR-Acc as the baseline as it is the baseline with the best average performance and efficiency in our work’s experimental results.
>
> | **1.** | SFT | STaR-Acc | AdaSTaR |
> |---|---|---|---|
> | Accuracy ($\uparrow$) | 80.0 | 88.0 | 89.0 |
> | PFLOPs ($\downarrow$) | 248.2 | 509.1 | 65.2 |
>
> | **2.** | SFT | STaR-Acc | AdaSTaR |
> |---|---|---|---|
> | Accuracy ($\uparrow$) | 71.2 | 84.1 | 84.3 |
> | PFLOPs ($\downarrow$) | 196.4 | 223.7 | 25.25 |
>
> | **3.** | SFT | STaR-Acc | AdaSTaR |
> |---|---|---|---|
> | Accuracy ($\uparrow$) | 42.8 | 68.2 | 68.4 |
> | PFLOPs ($\downarrow$) | 10.55 | 186.15 | 97.4 |
>
> This suggests that AdaSTaR's adaptive sampling benefits persist even in generalization and data mixture set-ups.
>
>
> ### **[W2] Early Stopping**
>
> This is a natural training behaviour of deep learning systems. The early-stopped iterations are very much akin to early-stopping epochs for SFT, in the sense that after that point the model overfits and accuracy tends to decrease. Practitioners can apply standard early stopping techniques [1] to capture these efficiency gains in practice.
>
> For example, training SVAMP with Gemma 7B shows clear overfitting after iteration 14 (peak accuracy 89.5%), with subsequent performance declining to 80.0% by iteration 21:
>
> | iter 1 | iter 2 | iter 3 | iter 4 | iter 5 | iter 6 | iter 7 | iter 8 | iter 9 | iter 10 | iter 11 | iter 12 | iter 13 | iter 14 | iter 15 | iter 16 | iter 17 | iter 18 | iter 19 | iter 20 | iter 21 |
> |--------|--------|--------|--------|--------|--------|--------|--------|--------|---------|---------|---------|---------|---------|---------|---------|---------|---------|---------|---------|---------|
> | 0.565  | 0.625  | 0.695  | 0.685  | 0.74   | 0.755  | 0.815  | 0.79   | 0.82   | 0.845   | 0.845   | 0.855   | 0.89    | 0.895   | 0.88    | 0.86    | 0.875   | 0.865   | 0.835   | 0.755   | 0.8     |
>
> For instance, "stop after two non-improving iterations" [1] would terminate at iteration 16 (14+2) with a final 89.5% accuracy, capturing efficiency gains. While AdaSTaR does not offer perfect prediction on when to early-stop, we do not believe that this is a weakness relative to related works: [2, 3, 4, 5, 6, 7, 10].
>
>
> ### **[Q2] Generalization to Unseen Test Data**
>
> We would like to clarify that AdaSTaR splits the training and test set to ensure that the test set is not seen during the training stage. You can feel free to view **Appendix E Further Details on Experimental Configuration and Setting**’s **Table 4**, which explicitly shows this.
>
> ### **[Q3] How are FLOPs Computed?**
>
> We will expand upon lines **259 - 261**:
>
> > We use FLOPs as our computational cost metric as memory usage remains approximately constant across methods. FLOPs are computed empirically following Kaplan et al. (2020), Sardana et al. (2024).
>
> We follow Kaplan et al. (2020) [8], Sardana et al. (2024) [9] to compute total flops as:
>
> $FLOPs := 6ND_{tr} + 2ND_{inf}$, where $N$ is the parameter size, and $D_{tr}$, and $D_{inf}$ are the total token count for training and inference respectively. Factor of 6 accounts for forward pass ($2N$) + backward pass ($4N$) in training. Factor of 2 accounts for forward pass only during inference
>
>
> ### **Reference**
>
> [1] Bengio, Yoshua. "Practical recommendations for gradient-based training of deep architectures." Neural networks 2012.
>
> [2] Eric Zelikman, Yuhuai Wu, Jesse Mu, and Noah Goodman. “STaR: Bootstrapping Reasoning With Reasoning”, NeurIPS 2022.
>
> [3] Xiangyu Peng, Congying Xia, Xinyi Yang, Caiming Xiong, Chien-Sheng Wu, and Chen Xing. “ReGenesis: LLMs can Grow into Reasoning Generalists via Self-Improvement”, ICLR 2025 (Oral).
>
> [4] Arian Hosseini, Xingdi Yuan, Nikolay Malkin, Aaron Courville, Alessandro Sordoni, and Rishab Agarwal. “V-STaR: Training Verifiers for Self-Taught Reasoners”, COLM 2024.
>
> [5] Richard Yuanzhe Pang, Weizhe Yuan, He He, Kyunghyun Cho, Sainbayar Sukhbaatar, and Jason E Weston. “Iterative Reasoning Preference Optimization”, NeurIPS 2024.
>
> [6] Weihao Zeng, Yuzhen Huang, Lulu Zhao, Yijun Wang, Zifei Shan, and Junxian He. “B-STaR: Monitoring and Balancing Exploration and Exploitation in Self-Taught Reasoners”, ICLR 2025.
>
> [7] Zheng Yuan, Hongyi Yuan, Chengpeng Li, Guanting Dong, Keming Lu, Chuanqi Tan, Chang Zhou, and Jingren Zhou. “Scaling Relationship on Learning Mathematical Reasoning with Large Language Models”, Arxiv 2023.
>
> [8] Jared Kaplan, Sam McCandlish, Tom Henighan, Tom B Brown, Benjamin Chess, Rewon Child, Scott Gray, Alec Radford, Jeffrey Wu, and Dario Amodei. “Scaling Laws for Neural Language Models”, Arxiv 2020.
>
> [9] Nikhil Sardana, Jacob Portes, Sasha Doubov, and Jonathan Frankle. “Beyond Chinchilla-Optimal: Accounting for Inference in Language Model Scaling Laws.” ICML 2024.
>
> [10] Avi Singh, John D Co-Reyes, Rishabh Agarwal, Ankesh Anand, Piyush Patil, Xavier Garcia, Peter J Liu, James Harrison, Jaehoon Lee, Kelvin Xu, Aaron T Parisi, Abhishek Kumar, Alexander A Alemi, Alex Rizkowsky, Azade Nova, Ben Adlam, Bernd Bohnet, Gamaleldin Fathy Elsayed, Hanie Sedghi, Igor Mordatch, Isabelle Simpson, Izzeddin Gur, Jasper Snoek, Jeffrey Pennington, Jiri Hron, Kathleen Kenealy, Kevin Swersky, Kshiteej Mahajan, Laura A Culp, Lechao Xiao, Maxwell Bileschi, Noah Constant, Roman Novak, Rosanne Liu, Tris Warkentin, Yamini Bansal, Ethan Dyer, Behnam Neyshabur, Jascha Sohl-Dickstein, and Noah Fiedel. “Beyond Human Data: Scaling Self-Training for Problem-Solving with Language Models”, TMLR 2024.
>
> [11] Teknium. “OpenHermes-2.5”, HuggingFace 2023.
>
> [12] Hugging Face Smol Models Research. “SmolTalk v2”, HuggingFace 2025.
>
> [13] Beijing Academy of Artificial Intelligence. “Infinity-Instruct”, HuggingFace 2024.
>
> [14] Niklas Muennighoff, Thomas Wang, Lintang Sutawika, Adam Roberts, Stella Biderman, Teven Le Scao, M Saiful Bari, Sheng Shen, Zheng-Xin Yong, Hailey Schoelkopf, Xiangru Tang, Dragomir Radev, Alham Fikri Aji, Khalid Almubarak, Samuel Albanie, Zaid Alyafeai, Albert Webson, Edward Raff, Colin Raffel. "Crosslingual generalization through multitask finetuning.", ACL 2023.
>
> [15] Jason Wei, Maarten Bosma, Vincent Y. Zhao, Kelvin Guu, Adams Wei Yu, Brian Lester, Nan Du, Andrew M. Dai, Quoc V. Le. “Finetuned Language Models are Zero-Shot Learners”, ICLR 2022 (Oral).

---

> ### Comment · Reviewer_2JDr · 2025-08-02
> **Response**
>
> Thank you for your response! The additional experiments strengthen the paper. But I still think AdaSTaR would need to pair with some ways of prediction on when to early-stop, in order to demonstrate its actual speed-up in a real-world setting. I decide to maintain my score.

---

> > ### Author Response · Authors · 2025-08-04
> > **Discussion Response**
> >
> > Thank you for your thoughtful engagement with our work. We believe that we have provided meaningful evidence regarding **[W1], [W2], [Q1], [Q2], [Q3]** above. As mentioned above, we believe common early-stopping mechanisms like "stop after two non-improving iterations" [1] will help capture most of the efficiency gains. It is not possible to encapsulate every possible improvement in a *single stand-alone* paper, and we believe that the current form is a significant contribution from existing works. Precise early-stopping mechanism is best left to a future work.
> >
> > We are continuously working on improving the updated paper based on your feedback.

---

### Official Review · Reviewer_suRc · 2025-07-01

**Clarity:** 3
**Significance:** 3
**Originality:** 2
**Rating:** 5
**Confidence:** 2

**Summary:**

This work proposes an adaptive finetuning method that prioritizes training data base on a so-called win statistic. This improves training efficiency over random sampling, as not every chain of thought is equally useful in downstream tasks, and some are even overly complicated yet lead to the correct outcome.

The authors show that STaR typically re-balances the training data, possibly inadvertently, and that the increased diversity can degrade quality of training CoTs.

The approach therefore proposes the following:
- tracking win statistics: how often we got the correct answer out of K CoT samples the last time this item was sampled
- using these statistics to sort a heap of datapoints by difficulty,
- training on the samples that were last sampled at more recent iterations, or deemed more difficult

This results in considerable training time improvements and/or an increase in performance over variations of STaR.

**Questions:**

1. When are some of the heuristic approximations a good choice, and when might they pose a risk?
2. How brittle or robust is this procedure compared to its (often simpler) alternatives? More insight would help convince me that this work is a useful procedure, and not just a well-tuned modification of STaR.
3. Iiuc the downside of the difficulty criterion is that it may still oversample too-difficult examples. This is currently countered by using alpha as guide for sampling. Can this have inadvertent effects, e.g. sampling near-impossible datapoints?

**Ethical Concerns:**

["NO or VERY MINOR ethics concerns only"]

**Final Justification:**

Authors adequately addressed my concerns and promised several improvements.

I believe the paper is technically solid, shows good improvement over the previous methods. It could be more clearly written, especially on what sets it apart from previous work. I therefore maintain my rating.

**Limitations:**

Yes

**Paper Formatting Concerns:**

None.

**Quality:**

3

**Strengths And Weaknesses:**

Strengths:
- Ability to cut training time in half yet get the same performance is quite compelling. This result is consistent, shown (with varying success but success nevertheless) on multiple datasets.
- Pseudocode makes it fairly easy to re-implement this method, if one understands the heuristic choices in section 3.1 well.
- Clear takeaways based on observations that are useful for practitioners using StaR or related methods.
- The empirical heap visualizations in the Appendix show the training procedure and evolution of priorities very well.

Weaknesses
- Some choices are heuristics, for example using the most recent estimate of the K-sample estimate of p_i^t relies n the assumption that the difference between the timesteps at which data was last sampled is not too large. This may not always be true in practice.
- The difficulty estimate similarly relies on an intuition that making a wrong prediction corresponds to difficulty of the sample. In some cases, datapoints may be impossible to solve for current model capacity, yet one correct verification means these will often be sampled as well iiuc. Do you consider this a problem?
- Combining model accumulation and adaptive data sampling through alpha makes the procedure potentially more sensitive to noise/outliers. Do you have any studies on variance across runs, or insights on when the system breaks?

---

> ### Author Rebuttal · Authors · 2025-07-30
>
> Thank you for your thoughtful and highly constructive comments. We are especially encouraged to see that you recognize:
>
> -   **Clear Takeaways, and Impact**
> 	-   “Clear takeaways … useful for practitioners” (`suRc`)
> -   **Consistently Strong Performance**
> 	-   “performance is quite compelling … result is consistent” (`suRc`)
> -   **Consistently Strong Efficiency Gains**
> 	-   “considerable training time improvements” (`suRc`)
> -   **Straight-forward Implementation**
> 	-   “fairly easy to re-implement” (`suRc`)
> -   **Clear Communication**
> 	-   “visualizations … show the training procedure and evolution of priorities very well” (`suRc`)
>
> We happily address the weaknesses **[W]** and questions **[Q]** you bring-forth in detail below. The following discussion has helped us improve parts of the updated version of our paper. Please feel free to share any further questions or concerns, as we welcome continued discussion for the broader research community.
>
> ### **[W1] $t - \tilde{t}_i$ May Be Large**
>
> We acknowledge that, in cases where the labeled train set $\mathcal{D}$ is large, the last sampled iteration may be large. However, we believe AdaSTaR remains effective even with larger $t - \tilde{t}_i$ for two reasons:
>
> 1.  Large post-training datasets typically comprise multiple sub-datasets [1, 2, 3, 4, 5] in practice, allowing AdaSTaR to be applied to each subset independently, keeping $t - \tilde{t}_i$ manageable.
>
> 2.  Our empirical evaluation (**Appendix E**, **Table 4**) across training sets of 800 to 10,000 observations shows no degradation or efficiency deterioration as dataset size increases. Contrarily, we see the opposite: consider our largest train set, ANLI. Across Llama 3.2 3B and Gemma 7B as pre-trained models, our results show consistent 1st place accuracy, and PFLOPs reduction of $\downarrow 62\%$ and $\downarrow 85.3\%$, respectively (**Table 1, 6**). This is a greater improvement than the average of $\downarrow 58.6\%$.
>
> Setting $t - \tilde{t}_i = 0$ would require significantly more inference compute, contradicting AdaSTaR's efficiency objective. $t - \tilde{t}_i = 0$ entails sampling each $i$, every iteration $t$. Nevertheless, we acknowledge that future works that choose to focus on maximizing peak accuracy over efficiency have the option to obtain statistics at $t$.
>
>
> ### **[W2] Win Rate = Difficulty?**
>
> We use win rate $w_i$ (**line 179**)​ as a difficulty proxy because it directly reflects how challenging each sample is for the specific pre-trained model being fine-tuned.
>
> > In some cases, datapoints may be impossible to solve for current model capacity, yet one correct verification means these will often be sampled as well iiuc. Do you consider this a problem?
>
> Regarding samples that are impossible for current model capacity: this concern motivates our design choice to prioritize the last sampled iteration $\tilde{t}$ in the HieMinHeap (**Equation 1**). By ordering samples primarily by $\tilde{t}$ the algorithm naturally cycles through different observations regardless of their difficulty, preventing it from getting stuck on any single challenging example.
>
> Furthermore, going one step further, to ensure that the algorithm does not get stuck on a near-impossible observation we make the following design choice in **lines 168 - 170**:
> > We use the last sampled iteration rather than the last trained iteration because prioritizing based on training can cause the system to repeatedly attempt difficult examples it cannot yet solve, particularly when the model is weak, early in training.
>
> In practice, we find that this design prevents the pathological behavior you describe while still appropriately prioritizing challenging samples that drive learning progress.
>
>
> ### **[W3] Robustness and Stability**
>
> Due to the high compute (on-policy reinforcement learning-like) nature of STaR approaches, having numerous runs was out of our computational budget. While multiple runs would be ideal, our experimental design prioritizes reproducibility and includes sufficient evidence of stability:
>
> 1.  As mentioned in **Appendix E**:
> > All training is done on the same arbitrary seed value of 10. This value has never been changed.
> 2.  Evaluation is done using greedy decoding (**Table 3**).
> 3.  In the paper, no instability or catastrophic failures can be observed across 12 experimental settings spanning three pre-trained model families (**Table 1, 5, 6**; **Figure 4**)
>
> We suspect that baselines and related works do not include numerous runs for variance or confidence interval as there is not a meaningful level of training instability. Specifically, [6, 7, 8, 9, 10, 11] do not report variance or confidence intervals, so we follow their evaluation protocol for fair comparison
>
> ### **[Q1] Heuristic Choice**
>
> We believe that our main heuristic choice comes from $f(\alpha) := \alpha^2$ where we discuss its underlying design choice in **lines 232 - 238**, and the footnote on **Page 6**. We acknowledge that this is a heuristic choice as we were unable to do an extensive empirical search on potential $f(\alpha)$.
>
> Nevertheless, we have put our best effort to point this out and discuss this in the paper. Consider **Appendix J**, **lines 780 - 783**:
>
> > Third, similar to other adaptive methods such as Adam (Kingma and Ba, 2015) and AdaGrad (Duchi et al., 2011), AdaSTaR introduces a new hyperparameter $f(\alpha) := \alpha^2$. A more granular tuning is deferred to future work. It is anticipated that such tuning could lead to further enhancements in AdaSTaR’s performance and efficiency.
>
> **When is $f(\alpha) := \alpha^2$ a good choice?** Our results show consistent improvements across 7 baselines and 3 model families despite minimal hyperparameter search, suggesting this choice is robust as a starting point. The quadratic form provides smooth curriculum transitions with monotonically increasing behavior ($f'(\alpha) = 2\alpha \geq 0$ for $\alpha \geq 0$) while maintaining stability across different architectures.
>
>
> **Potential risks.** The main risk is suboptimality; a more thorough search over $f(\alpha)$ could yield better performance. However, we observed stable training dynamics across all experiments, indicating low risk of training instability.
>
> Finally, we acknowledge that including an intuitive explanation for future researchers and practitioners’ $f(\alpha)$ choice is important. We are planning to add the following part to the paper:
>
> > $f(\alpha)$ should be set such that the range is $f(\alpha) \in [0, 1]$. A value of 0 corresponds to encouraging easy observations, while 1 corresponds to completely turning off the curriculum mechanism.
>
> ### **[Q2] Well Tuned STaR?; and Robustness**
>
> We respectfully disagree that AdaSTaR is a well-tuned version of STaR for the following three **(1, 2, 3)** reasons.
>
> **(1)** First, we note in **lines 244 - 245**:
> > For fairness, we optimize hyperparameters using the original STaR and apply them consistently across all methods
>
> That is, we have already given a common hyperparameter search advantage to the original STaR baseline.
>
> **(2) Algorithmic novelty.** AdaSTaR introduces fundamentally new components: (i) online difficulty statistics ($\tilde{t}$, $w$) computed with negligible overhead, and (ii) adaptive curriculum scheduling via $f(\alpha)$. These are not hyperparameter tuning choices but algorithmic innovations absent from prior work.
>
> **(3)** Lastly, as mentioned in our response to **[Q1]**, our single hyperparameter $f(\alpha) := \alpha^2$ was chosen without extensive search, yet consistently outperforms 7 baselines across 3 model families. This suggests robustness rather than cherry-picking.
>
> We believe that the robustness part of this question is intimately related to our response to **[W3]**. We would like to further note that the aforementioned [7, 8, 9, 10, 11] in **[W3]** are significantly more complex versions of STaR.
>
> ### **[Q3] Sampling Near-impossible Observations?**
>
> Please refer to our response to **[W2]**.
>
> ### **Reference**
> [1] Teknium. “OpenHermes-2.5”, HuggingFace 2023.
>
> [2] Hugging Face Smol Models Research. “SmolTalk v2”, HuggingFace 2025.
>
> [3] Beijing Academy of Artificial Intelligence. “Infinity-Instruct”, HuggingFace 2024.
>
> [4] Niklas Muennighoff, Thomas Wang, Lintang Sutawika, Adam Roberts, Stella Biderman, Teven Le Scao, M Saiful Bari, Sheng Shen, Zheng-Xin Yong, Hailey Schoelkopf, Xiangru Tang, Dragomir Radev, Alham Fikri Aji, Khalid Almubarak, Samuel Albanie, Zaid Alyafeai, Albert Webson, Edward Raff, Colin Raffel. "Crosslingual generalization through multitask finetuning.", ACL 2023.
>
> [5] Jason Wei, Maarten Bosma, Vincent Y. Zhao, Kelvin Guu, Adams Wei Yu, Brian Lester, Nan Du, Andrew M. Dai, Quoc V. Le. “Finetuned Language Models are Zero-Shot Learners”, ICLR 2022 (Oral).
>
> [6] Eric Zelikman, Yuhuai Wu, Jesse Mu, and Noah Goodman. “STaR: Bootstrapping Reasoning With Reasoning”, NeurIPS 2022.
>
> [7] Xiangyu Peng, Congying Xia, Xinyi Yang, Caiming Xiong, Chien-Sheng Wu, and Chen Xing. “ReGenesis: LLMs can Grow into Reasoning Generalists via Self-Improvement”, ICLR 2025 (Oral).
>
> [8] Arian Hosseini, Xingdi Yuan, Nikolay Malkin, Aaron Courville, Alessandro Sordoni, and Rishab Agarwal. “V-STaR: Training Verifiers for Self-Taught Reasoners”, COLM 2024.
>
> [9] Richard Yuanzhe Pang, Weizhe Yuan, He He, Kyunghyun Cho, Sainbayar Sukhbaatar, and Jason E Weston. “Iterative Reasoning Preference Optimization”, NeurIPS 2024.
>
> [10] Weihao Zeng, Yuzhen Huang, Lulu Zhao, Yijun Wang, Zifei Shan, and Junxian He. “B-STaR: Monitoring and Balancing Exploration and Exploitation in Self-Taught Reasoners”, ICLR 2025.
>
> [11] Zheng Yuan, Hongyi Yuan, Chengpeng Li, Guanting Dong, Keming Lu, Chuanqi Tan, Chang Zhou, and Jingren Zhou. “Scaling Relationship on Learning Mathematical Reasoning with Large Language Models”, Arxiv 2023.

---

> > ### Comment · Reviewer_suRc · 2025-08-02
> > **Thanks for the detailed response**
> >
> > - W1: thanks for the clarification and pointing me to the results in the Appendix. I also think reducing or constraining the difference can be counterproductive.
> > - W2: I think I understand how using the sampled iteration can increase diversity and avoid getting stuck. But I don't see how it completely alleviates the problem - there may still be datapoints in the batch that are too difficult to solve, and they will keep appearing as long as win rate is a criterion for sampling.
> > - W3. Greedy decoding with fixed seed alleviates this concern somewhat, but I think it's problematic that we do not know whether the accuracy differences (<= 2% to the next best baseline in Tab 1) are meaningful. That said, the efficiency gains (PFLOPs) are of course substantial, I doubt these would change considerably.
> >
> > - Q1: Thanks, this is clear. Acknowledging this is useful.
> > - Q2: I think the novelty can be a downside if the method contains more moving parts. But I acknowledge that you've attempted to give the Star baseline equal tuning time, and outperform it.
> > - Q3: noted.
> >
> > Note that I did not say it is straightforward to reimplement this method. *If one understands the heuristics well*, I think it's easy to do so with the pseudocode in hand. Rephrasing is fine, but don't change my statements please.
> >
> > I don't understand every heuristic well. They are described in plain text in section 3, and I think I understand some of them, but I could easily miss one if I wanted to implement them. It would be beneficial if the most important design choices compared to STaR + corresponding intuition are listed somewhere (perhaps Appendix).

---

> > > ### Author Response · Authors · 2025-08-04
> > > **Discussion Response (Part I)**
> > >
> > > Thank you for the follow-up, and sharing your time contributing to the research community. We appreciate your constructive comments. The following is our response.
> > >
> > > ### **[W2] Response: Sampling Challenging Observations**
> > >
> > > We appreciate the reviewer’s insight that datapoints with very low win-rates (e.g., $w_i = 0$) will eventually be re-sampled. This behavior is indeed intentional and aligns with our design, which we elaborate below.
> > >
> > > **Equation (1)** first sorts by the last-sampled iteration $\tilde{t}$; the win-rate $w_i$ is only a secondary tie-breaker among observations with the same $\tilde{t}$ value. Therefore, difficulty *never accelerates* re-selection: once a hard sample is drawn at iteration $t$, it cannot be revisited until *all* observations with more recent $\tilde t$ values have been processed. This introduces a natural delay, giving the model time to improve before facing the same example again.
> > >
> > > We visualize this behavior in **Appendix G**. For example, even the most difficult sample (e.g., $w_i = 0$ at $t = 1$) are only re-sampled at later iterations (e.g., $t = 11$ in CQA and $t = 12$ in ANLI), well after other observations have been sampled. This way, the model avoids repeatedly failing on hard examples too early, but still revisits them later after gaining more experience.
> > >
> > > By combining recency-based scheduling with difficulty-aware tie-breaking, it keeps the model from getting stuck on hard cases while guaranteeing that every example is eventually revisited, yielding the right mix of *diversity* and *steadily increasing difficulty* for continual curriculum learning.
> > >
> > > ### **[W3] Response: Meaningful Accuracy and PFLOPs Efficiency Gains**
> > >
> > >
> > > > but I think it's problematic that we do not know whether the accuracy differences (<= 2% to the next best baseline in Tab 1) are meaningful
> > >
> > > First, we acknowledge that the PFLOPs efficiency gains are more pronounced than the accuracy gains. While the accuracy gains are not as large as the PFLOPs gains, we believe they are still *meaningful* for practitioners for the following reasonings:
> > >
> > > 1.  **Consistency Across Models and Datasets:** While the absolute accuracy gains are modest, AdaSTaR achieves the best accuracy on *every* single task, across **6x** datasets and **3x** different pre-trained base models.
> > >
> > > 2.  **Comprehensive Baseline Coverage:** We compare against **10** baselines (**7x** STaR-based and **3x** SFT-based methods). To our knowledge, this covers all reasonable alternatives used in related work.
> > >
> > > This consistency is unlikely to be due to chance and suggests robustness across problem settings.
> > >
> > > Following the precedent in prior work [1, 2, 3, 4, 5, 6, 7, 8, 9], statistical significance tests are not commonly performed in this area. While we agree with your point, this issue is not limited to our work; we adopted the standard evaluation protocol and reporting style established by prior literature to ensure fair comparison. That said, if additional time is available before the camera-ready version, we would be happy to incorporate more rigorous statistical reporting to further strengthen the empirical claims.

---

> > > ### Author Response · Authors · 2025-08-04
> > > **Discussion Response (Part II)**
> > >
> > > ### **[Additional Discussion: STaR vs. AdaSTaR]**
> > >
> > > > If one understands the heuristics well, I think it's easy to do so with the pseudocode in hand.
> > >
> > > Thank you for pointing this out. This was a misunderstanding on our part. We do not believe that we are able to edit our original rebuttal response, but if given a chance we will remove the statement: “Straight-forward Implementation”.
> > >
> > > We will spend the remaining time trying to add a section in the **Appendix**, making the difference against STaR clear. Nevertheless, we would like to share the effort we have put in the existing paper so far to make the design choices and implementation clear. We believe that we have not omitted any information in the main paper which makes it challenging to re-implement. The pseudocode (**Algorithm 1**) is a near exact implementation of the source code.
> > >
> > > 1.  **High Level Visualization**:  **Figure 2**’s caption indicates that the high-level diagram of STaR vs. AdaSTaR can be understood as without the green part vs. with the green part.
> > > 2.  **Algorithm 1’s Color-coding**: As mentioned in **line 163**, STaR vs. AdaSTaR can be intuitively compared via: the black parts corresponding to STaR against the green parts that correspond to AdaSTaR.
> > > 	3. Moreover, in **Algorithm 1**, we clearly comment the parts corresponding to the *diversity* design choice as “AdaD (\S 3.1}; lines 1-14), and the *curriculum* design choice as “AdaC (\S 3.2; lines 15-19)”
> > > 3.  **HieMinHeap’s Mechanism**: clearly explained in **Equation (1)**
> > > 4.  **Writing Flow: Mapping of Motivation Section 2.2 → Design Choice**
> > > 	5.  2.2 Subsection 1 → Section 3.1
> > > 	6.  2.2 Subsection 2 → Section 3.2
> > >
> > > | Motivation Section 2.2's sub-section | Method Design Choice Section | Introduced Statistic | Introduced Algorithmic Design Choice |
> > > |---------------------------------------|------------------------------|---------------------|-------------------------------------|
> > > | STaR's data sampling induces persistent inefficient imbalance in training data (**lines 122 - 140**) | Section 3.1 Adaptive Data Sampling for Diversity (**lines 166 - 216**) | $\tilde{t}_i$, $w_i$ | Priority queue sample based on $\tilde{t}_i$ and $w_i$ |
> > > | Prioritizing harder examples for diversity elevates false positives, motivating curriculum based regularization. (**lines 141 - 159**) | Section 3.2 Adaptive Data Sampling for Curriculum (**lines 218 - 231**) | NA | Mix easy observations by modifying the pop/push mechanism, depending on the model's underlying strength |
> > >
> > > We are continuously working on improving the updated paper based on your feedback. Please feel free to share if we missed anything or misunderstood your discussion points.
> > >
> > >
> > > ### **Reference**
> > >
> > > [1] Eric Zelikman, Yuhuai Wu, Jesse Mu, and Noah Goodman. “STaR: Bootstrapping Reasoning With Reasoning”, NeurIPS 2022.
> > >
> > > [2] Xiangyu Peng, Congying Xia, Xinyi Yang, Caiming Xiong, Chien-Sheng Wu, and Chen Xing. “ReGenesis: LLMs can Grow into Reasoning Generalists via Self-Improvement”, ICLR 2025 (Oral).
> > >
> > > [3] Arian Hosseini, Xingdi Yuan, Nikolay Malkin, Aaron Courville, Alessandro Sordoni, and Rishab Agarwal. “V-STaR: Training Verifiers for Self-Taught Reasoners”, COLM 2024.
> > >
> > > [4] Richard Yuanzhe Pang, Weizhe Yuan, He He, Kyunghyun Cho, Sainbayar Sukhbaatar, and Jason E Weston. “Iterative Reasoning Preference Optimization”, NeurIPS 2024.
> > >
> > > [5] Weihao Zeng, Yuzhen Huang, Lulu Zhao, Yijun Wang, Zifei Shan, and Junxian He. “B-STaR: Monitoring and Balancing Exploration and Exploitation in Self-Taught Reasoners”, ICLR 2025.
> > >
> > > [6] Zheng Yuan, Hongyi Yuan, Chengpeng Li, Guanting Dong, Keming Lu, Chuanqi Tan, Chang Zhou, and Jingren Zhou. “Scaling Relationship on Learning Mathematical Reasoning with Large Language Models”, Arxiv 2023.
> > >
> > > [7] Jared Kaplan, Sam McCandlish, Tom Henighan, Tom B Brown, Benjamin Chess, Rewon Child, Scott Gray, Alec Radford, Jeffrey Wu, and Dario Amodei. “Scaling Laws for Neural Language Models”, Arxiv 2020.
> > >
> > > [8] Nikhil Sardana, Jacob Portes, Sasha Doubov, and Jonathan Frankle. “Beyond Chinchilla-Optimal: Accounting for Inference in Language Model Scaling Laws.” ICML 2024.
> > >
> > > [9] Avi Singh, John D Co-Reyes, Rishabh Agarwal, Ankesh Anand, Piyush Patil, Xavier Garcia, Peter J Liu, James Harrison, Jaehoon Lee, Kelvin Xu, Aaron T Parisi, Abhishek Kumar, Alexander A Alemi, Alex Rizkowsky, Azade Nova, Ben Adlam, Bernd Bohnet, Gamaleldin Fathy Elsayed, Hanie Sedghi, Igor Mordatch, Isabelle Simpson, Izzeddin Gur, Jasper Snoek, Jeffrey Pennington, Jiri Hron, Kathleen Kenealy, Kevin Swersky, Kshiteej Mahajan, Laura A Culp, Lechao Xiao, Maxwell Bileschi, Noah Constant, Roman Novak, Rosanne Liu, Tris Warkentin, Yamini Bansal, Ethan Dyer, Behnam Neyshabur, Jascha Sohl-Dickstein, and Noah Fiedel. “Beyond Human Data: Scaling Self-Training for Problem-Solving with Language Models”, TMLR 2024.

---

> > > > ### Comment · Reviewer_suRc · 2025-08-07
> > > >
> > > > Thanks for the additional results and clarifications. I think they will strengthen the paper and make it easier for others to compare your work to existing work.

---

### Author Response · Authors · 2025-08-07
**Global Discussion Response: Additional Experimental Results**

We appreciate all reviewers and `AC` engagement. As the discussion period comes to an end, we would like to direct other reviewers' attention to additional experiments that we were able to run during the rebuttal phase. We are grateful to reviewer `2JDr` whom suggested additional diverse experimental set-ups: **(1)** Transfer generalization evaluations, **(2)** Data mixture training.

The experiment protocol here is:

**1.**  Train: GSM8K → Test: SVAMP

**2.**  Train: GSM8K + SVAMP → Test: GSMK + SVAMP

**3.**  Train: SVAMP + ARC-C → Test: GSM8K + CQA

The base model is Qwen 2.5 3B, and the train set is randomly shuffled. All other settings remain consistent with the original paper. We choose STaR-Acc as the baseline as it is the baseline with the best average performance and efficiency in our work’s experimental results.

| **1.** | SFT | STaR-Acc | AdaSTaR |
|---|---|---|---|
| Accuracy ($\uparrow$) | 80.0 | 88.0 | 89.0 |
| PFLOPs ($\downarrow$) | 248.2 | 509.1 | 65.2 |

| **2.** | SFT | STaR-Acc | AdaSTaR |
|---|---|---|---|
| Accuracy ($\uparrow$) | 71.2 | 84.1 | 84.3 |
| PFLOPs ($\downarrow$) | 196.4 | 223.7 | 25.25 |

| **3.** | SFT | STaR-Acc | AdaSTaR |
|---|---|---|---|
| Accuracy ($\uparrow$) | 42.8 | 68.2 | 68.4 |
| PFLOPs ($\downarrow$) | 10.55 | 186.15 | 97.4 |


The above reports **all** experiments ran during the rebuttal phase.

Please refer to our full response to `2JDr` (**[W1 & Q1] Transfer Generalization and Mixture of Datasets**) for extended details and discussion.

---

### Decision · Program_Chairs · 2025-09-17

**Decision:**

Accept (poster)

**Comment:**

This paper introduces AdaSTaR, an adaptive data sampling method to improve the efficiency of STaR. The core contribution is replacing STaR's inefficient random sampling, which often over-trains on easy examples while under-training on difficult ones, with two adaptive sampling strategies: one that promotes diversity by prioritizing less recently sampled data points, and another that creates a curriculum by dynamically adjusting data difficulty to match the model's current capabilities.

The primary strength of this work is its compelling and consistent empirical result; it successfully addresses a well-known "pain point" in STaR by substantially improving training efficiency and model performance across a broad range of datasets and models. The proposed method is also simple and lightweight, making it easy to implement within existing STaR frameworks. The main weaknesses were the heuristic nature of some key design choices and the practical realization of the efficiency gains is dependent on an effective early-stopping strategy, which the paper does not provide.

During the rebuttal period, the authors were highly responsive and effectively addressed most of the reviewers' concerns. New experiments added also addressed the concern about the merits of Adastar in transferability and generalizability. Overall, the paper is sound, and the demonstrated improvements in training efficiency are significant. The problem of inefficiently sampling easy problems is a known limitation of STaR, and this paper provides a practical and effective solution to this pain point. Initial concerns about the method's generalizability were well-addressed during the rebuttal. Of the remaining weaknesses, for the heuristic nature of the curriculum, the AC believes that the substantial practical gains appear to outweigh the lack of a theoretical proof of optimality. The dependency on early stopping is a valid point, and while it is a general problem in the field, the authors are encouraged to discuss this limitation more explicitly. The merits of this work ultimately outweigh its flaws.